# Chaos is a Ladder: A New Theoretical Understanding of Contrastive Learning via Augmentation Overlap

**Yifei Wang**[1*]  **Qi Zhang**[2*]  **Yisen Wang**[3,4†]  **Jiansheng Yang**[1]  **Zhouchen Lin**[3,4,5]

[1] School of Mathematical Sciences, Peking University
[2] School of Computer Science and Engineering, Sun Yat-sen University
[3] Key Lab. of Machine Perception (MoE), School of Artificial Intelligence, Peking University
[4] Institute for Artificial Intelligence, Peking University
[5] Pazhou Lab, Guangzhou, 510330, China

## Abstract

Recently, contrastive learning has risen to be a promising approach for large-scale self-supervised learning. However, theoretical understanding of how it works is still unclear. In this paper, we propose a new guarantee on the downstream performance without resorting to the conditional independence assumption that is widely adopted in previous work but hardly holds in practice. Our new theory hinges on the insight that the support of different intra-class samples will become more overlapped under aggressive data augmentations, thus simply aligning the positive samples (augmented views of the same sample) could make contrastive learning cluster intra-class samples together. Based on this *augmentation overlap* perspective, theoretically, we obtain asymptotically closed bounds for downstream performance under weaker assumptions, and empirically, we propose an unsupervised model selection metric ARC that aligns well with downstream accuracy. Our theory suggests an alternative understanding of contrastive learning: the role of aligning positive samples is more like a surrogate task than an ultimate goal, and the overlapped augmented views (i.e., the chaos) create a ladder for contrastive learning to gradually learn class-separated representations. The code for computing ARC is available at https://github.com/zhangq327/ARC.

## 1 Introduction

Contrastive Learning (CL) emerges to be a promising paradigm for learning data representations without labeled data (Oord et al., 2018; Hjelm et al., 2019). Recently, it has achieved impressive results and gradually closed the gap between supervised and unsupervised learning, hopefully leading to a new era that resolves the hunger for labeled data in the deep learning field (He et al., 2020; Chen et al., 2020b; Wang et al., 2021). However, despite its intriguing empirical success, a theoretical understanding of how contrastive learning actually works in practice is still under-explored.

The general methodology of contrastive learning is quite simple, that is to maximize the similarity between augmented views of the same image (*a.k.a.* positive samples), and minimize the similarity between that of two random images (*a.k.a.* negative samples). Intuitively, it is *an instance discrimination task* (differing each image from others) instead of *a classification task* (clustering images from the same class together and differing with other classes). Nevertheless, as shown in Figure 1(a), CL representations are also class-separated. Therefore, understanding how the pretraining task (CL) and the downstream task (classification) interact plays a central role in both theoretical understandings and practical designings of contrastive methods.

Previously, Saunshi et al. (2019) and Lee et al. (2020) have tried to establish guarantees on the classification performance for self-supervised representations. However, their analysis relies heavily on the assumption that the two positive samples, as augmented views of the same image, are (nearly) conditionally independent on the class $y$. However, this is hardly practical as the augmented views are still strongly input-dependent (see Figure 1(b)). In fact, if the conditional independence is satisfied, the unsupervised task will become as informative as the supervised task, making this discussion

---

*Equal contribution. Qi Zhang's work was done during an internship at Peking University.
†Corresponding author: Yisen Wang (yisen.wang@pku.edu.cn).

Before Training

After Training

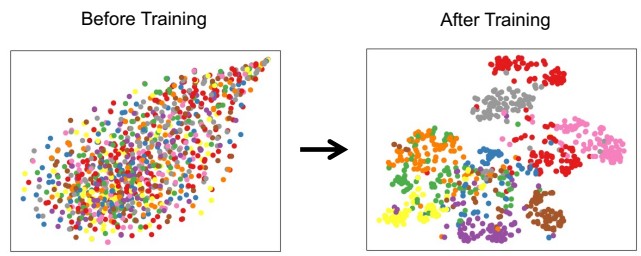

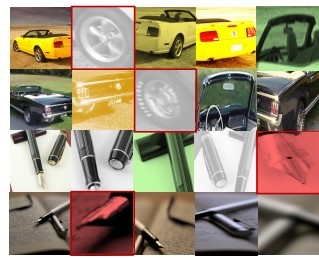

(a) Contrastive learning learns clustered features.

(b) Intra-class samples are more alike via augmented views.

Figure 1: (a) t-SNE visualization of representations before and after contrastive learning. Each point denotes a sample and its color denotes its class. (b) Applying aggressive data augmentations (Chen et al., 2020a) to four images from ImageNet (two are cars and two are pens). The 1st column shows the raw (center-cropped) images and the 2-5th colums show the augmented ones.

almost unnecessary. This motivates us to find more practical and weaker assumptions to understand how contrastive learning actually works (even without conditional independence). To achieve this, we need to re-examine the contrastive learning process. Previously, Wang & Isola (2020) show that CL objective involves two goals: alignment (for positive samples) and uniformity (for negative samples). Nevertheless, we show that there exist bad cases where the features could still have poor performance even with perfect alignment and uniformity. Thus, contradictory to the common belief of contrastive learning as learning invariance, we note that invariance alone is inadequate to learning useful representations for downstream tasks.

In this paper, we provide a novel understanding of contrastive learning that requires only practical and minimal assumptions, while also guarantee class-separated representations. Our core insight hinges on the observation that contrastive learning usually adopts much more aggressive data augmentations than that in supervised learning (He et al., 2020; Chen et al., 2020a). As shown in Figure 1(b), we notice that aggressive random cropping of two images can generate views that are very much alike that we could even hardly tell them apart, *e.g.*, the wheels of two different cars. In other words, there will be support overlap between different intra-class images through aggressively augmented views of them, a phenomenon we call *augmentation overlap*. Thus, the alignment of positive samples will also cluster all the intra-class samples together, and lead to class-separated representations. From our perspective, the role of data augmentation is to create a certain degree of "chaos" between intra-class samples, and the role of contrastive loss is to "climb the ladder of chaos", *i.e.*, the process that we gradually cluster intra-class samples by aligning positive samples.

Following this intuition, we develop a new theory for understanding the effectiveness of contrastive learning from the perspective of augmentation overlap. Specifically, we derive the upper and lower bounds for its downstream performance and show how the two bounds will asymptotically converge with our assumptions on augmentation overlap. Driven by this analysis, we further discuss how varying augmentation will affect the performance of contrastive learning from both synthetic and real-world datasets, and show that the results align well with our theory. In summary,

- We characterize the failure of the previous analysis of contrastive learning, and develop a new understanding through the augmentation overlap effect. Compared to existing theories on contrastive learning, ours can provide guidance to the practical designing of contrastive methods and evaluation metrics.
- We establish general guarantees (both upper and lower bounds) for the downstream performance without assumptions on conditional independence. And we further show how the two bounds could asymptotically converge under our less restrictive assumptions.
- We provide a quantitative discussion on the effect of augmentation strength, which verifies our theory from both theoretical and empirical aspects. Motivated by our theory, we further propose a new unsupervised evaluation metric for contrastive learning named ARC and show that it aligns well with downstream performance on real-world datasets.

## 2 RELATED WORK

**Contrastive Learning in Practice.** Contrastive self-supervised learning originates from a mutual information perspective of representation learning (Oord et al., 2018; Hjelm et al., 2019), and soon

becomes a general learning paradigm that contrasts between positive and negative pairs (He et al., 2020; Chen et al., 2020a). It is rapidly closing the performance gap between unsupervised and supervised learning on large-scale dataset like ImageNet (Chen et al., 2021), and outperforms supervised learning when combined with a few (*e.g.,* 10%) labels (Chen et al., 2020b). Several recent works show that similar performance could be achieved without negative samples by adopting certain training techniques (Grill et al., 2020; Chen & He, 2020).

**Understanding Contrastive Learning Objectives.** Both the original InfoNCE loss (Oord et al., 2018) and its InfoMax variants (Hjelm et al., 2019; Poole et al., 2019) are designed as variational estimates of the mutual information between inputs and representations, but these estimators are shown to have poor bias-variance trade-offs (Song & Ermon, 2020). Instead, Wang & Isola (2020) simply understand contrastive learning through the two terms in the InfoNCE loss: alignment of positive samples and uniformity of negative samples. However, as we show later, this perspective is also insufficient to explain the effectiveness of contrastive learning, and we should take the interplay between augmentation and alignment into consideration.

**Understanding Downstream Generalization.** Saunshi et al. (2019) propose the first theoretical guarantees by bridging the contrastive and classification objectives. Lee et al. (2020) further link the reconstruction-based objective to the downstream objective. However, both Saunshi et al. (2019) and Lee et al. (2020) rely on the unrealistic assumption that the positive samples are (nearly) conditionally independent. Huang et al. (2021) establish bounds by assuming a very small intra-class support diameter, which is also not practical. Besides, some also explore the information-theoretical perspectives for analyzing contrastive learning (Tian et al., 2020; Tsai et al., 2021; Tosh et al., 2020; 2021), though their mutual information assumptions are hard to verify. Recently, similar to our analysis, HaoChen et al. (2021) also study the augmentation graph and establish guarantees in terms of graph connectivity. Our work differs to theirs mainly in three aspects: 1) our analysis is applicable for the widely adopted InfoNCE and CE losses, while theirs is developed for their own spectral loss; 2) ours starts from the alignment and uniformity perspective while theirs starts from the matrix decomposition perspective; 3) our theory is empirically verified and inspires a useful evaluation metric for data augmentation, while their analysis focusing on minimizing the decomposition error is farther from the practical designing of positive and negative samples. In a nutshell, compared to previous discussions, our theory has a closer connection to the actual contrastive learning process, and we verify the feasibility of each assumption with empirical evidence.

## 3 LIMITATIONS OF PREVIOUS UNDERSTANDINGS

We begin by introducing the basic notations and common practice of contrastive learning in the image classification task. In general, it has two stages, unsupervised pretraining, and supervised finetuning. In the first stage, with $N$ unlabeled samples $\mathcal{D}_u = \{x_i\}_{i=1}^N$, we pretrain an encoder mapping from the $d$-dimensonal input space to a unit hypersphere $f \in \mathcal{F} : \mathbb{R}^d \to \mathbb{S}^{m-1}$ in the $m$-dimensional space. In the second stage, we evaluate the learned representations $z$ with the labeled data $\mathcal{D}_l = \{(x_i, y_i)\}$ where labels $y_i \in \{1, \dots, K\}$. Specifically, we fix the encoder and learn a linear classification head $g : \mathcal{R}^m \to \mathcal{R}^K$ on top from $\tilde{\mathcal{D}}_l = \{(z, y) | z = f(x) \in \mathcal{R}^m\}$.

**Contrastive Pretraining**. Taking a training example $x \in \mathcal{D}_u$, we draw its positive sample $x^+ = t(x)$ by applying a random data augmentation $t \sim \mathcal{T}$, and draw $M$ randomly augmented samples $\{x_i^-\}_{i=1}^M$ from $\mathcal{D}_u$ as its negative samples. Then, we can learn the encoder $f$ with the widely used InfoNCE loss (Oord et al., 2018)

$$\mathcal{L}_{\text{NCE}}(f) = \mathbb{E}_{p(x,x^+)} \mathbb{E}_{\{p(x_i^-)\}} \left[ -\log \frac{\exp(f(x)^\top f(x^+))}{\sum_{i=1}^M \exp(f(x)^\top f(x_i^-))} \right]. \tag{1}$$

Let $p(x)$ be the data distribution, $p(x, x^+)$ be the joint distribution of positive pairs, and we simply assume $p(x, x^+) = p(x^+, x)$ and $p(x) = \int p(x, x^+) dx^+, \forall x \in \mathbb{R}^d$ following Wang & Isola (2020).

**Linear Evaluation**. To evaluate the learned representations by contrastive learning, we usually adopt the Cross Entropy (CE) loss (Chen et al., 2020a) for a labeled pair $(x, y) \in \mathcal{D}_l$

$$\mathcal{L}_{\text{CE}}(f, g) = \mathbb{E}_{p(x,y)} \left[ -\log \frac{\exp\left(f(x)^\top w_y\right)}{\sum_{i=1}^K \exp\left(f(x)^\top w_i\right)} \right], \tag{2}$$

with a linear classifier $g(z) = Wz$ where $W = [w_1, w_2, \dots, w_K]$.

## 3.1 Existing Theoretical Assumptions and Their Limitations

As discussed above, there are some previous understandings on how contrastive learning yields good performance, and they mainly differ by their theoretical assumptions.

First, Wang & Isola (2020) interpret the first and second terms of the InfoNCE loss (Eq. 1) as they are aiming at the following two properties: 1) alignment (the nominator): positive samples $x, x^+$ has similar features, *i.e.*, $f(x) \approx f(x^+)$; 2) uniformity (the denominator): features are roughly uniformly distributed in the unit hypersphere $\mathbb{S}^{m-1}$. In particular, they show that InfoNCE can be minimized with 1) perfect alignment and 2) perfect uniformity. However, as we illustrate in Figure 2, the features could still have very poor downstream performance in the finite sample scenario. This issue can be described rigorously by the following proposition.

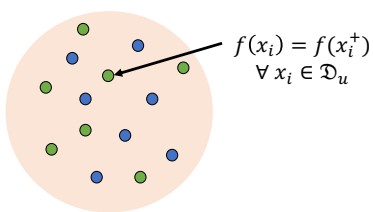

Figure 2: Contrastive learning may learn class inseparable features even with perfect aligned positive samples and uniform negative samples. Colors denote classes.

**Proposition 3.1** (Class-uniform Features Also Minimize the InfoNCE Loss). *For $N$ training examples of $K$ classes, consider the case when features $\{f(x_i)\}_{i=1}^N$ are randomly distributed in $\mathbb{S}^{m-1}$ with maximal uniformity (i.e., , minimizing the 2nd term of Eq. 1) while also satisfying $\forall x_i, x_i^+ \sim p(x, x^+), f(x_i) = f(x_i^+)$. Because we have these two properties, the InfoNCE loss achieves its minimum. However, the downstream classification accuracy is at most $1/K + \varepsilon$ and $\varepsilon$ is nearly zero when $N$ is large enough.*

Proofs can be found in Appendix A. This proposition indicates that the instance discrimination task (alignment + uniformity) alone cannot guarantee the learning of class-discriminative features as desired in the downsteam classification. Instead, Saunshi et al. (2019) and Lee et al. (2020) both establish the relationship between pretraining and classification objectives and provide guarantees for the downstream performance. In fact, the two works both assume the conditional independence of the two positive samples, *i.e.*, $p(x, x^+|y) = p(x|y)p(x^+|y)$. However, this assumption is too strong as it is hardly practical. As shown in Figure 1(b), augmented views from the same class are not actually independent as views from the same sample are more alike than that from other samples.

## 4 New Augmentation Overlap Theory for Contrastive Learning

The analysis above motivates us to find a *minimal* and *practical* assumption: 1) it is enough to guarantee good performance on downstream tasks; 2) it is less restrictive than the i.i.d. assumptions as in Saunshi et al. (2019) and Lee et al. (2020).

### 4.1 Gap Between Contrastive Learning and Downstream Classification

We start with an assumption on the label consistency between positive samples, that is, any pair of positive samples $(x, x^+)$ should belong to the same class.

**Assumption 4.1** (Label Consistency). *$\forall x, x^+ \sim p(x, x^+)$, we assume the labels are deterministic (one-hot) and consistent: $p(y|x) = p(y|x^+)$.*

This is a natural and minimal assumption that is likely to hold in practice. As shown in Figure 1(b), the widely adopted augmentations in contrastive learning (Chen et al., 2020a) like images cropping, color distortion, and horizontal flipping will hardly alter the belonging image classes.

With this minimal assumption, we can characterize the generalization gap between unsupervised and supervised learning risks. We first introduce the mean CE loss, $L_{\mathrm{CE}}^{\mu}(f) = \mathbb{E}_{p(x,y)}\left[-\log \frac{\exp\left(f(x)^{\top}\mu_y\right)}{\sum_{i=1}^{K}\exp(f(x)^{\top}\mu_i)}\right]$, where we use the classwise *mean representation* $\mu_k = \mathbb{E}_{p(x|y=k)}[f(x)]$ as the weight $w_k$ of the classifier $g$. It is easy to see that the mean CE loss upper bounds the CE loss, *i.e.*, $L_{\mathrm{CE}}^{\mu}(f) \geq \min_g \mathcal{L}_{\mathrm{CE}}(f, g)$ and Saunshi et al. (2019) showed that the mean classifier could achieve comparable performance to learned weights. Then, we have the following upper and lower bounds on the downstream risk (measured by mean CE loss).

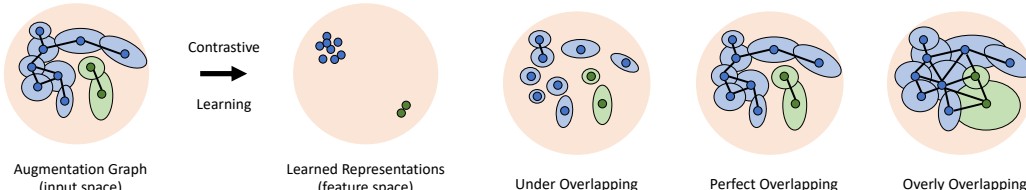

(a) Contrastive learning with an augmentation graph satisfying intra-class connectivity.

(b) Augmentation graph under increasing augmentation strengthes (left to right).

Figure 3: Illustrative examples of augmentation graphs, where each dot denotes a sample $x \in \mathcal{D}_u$ and its color denotes its class. The lighter disks denote the support of the positive samples $p(x^+|x)$. We draw a solid edge for each $\mathcal{T}$-connected pair.

**Theorem 4.2** (Guarantees for General Encoders). *If Assumption 4.1 holds, then, for any $f \in \mathcal{F}$, its downstream classification risk $\mathcal{L}_{\mathrm{CE}}^{\mu}(f)$ can be bounded by the contrastive learning risk $\mathcal{L}_{\mathrm{NCE}}(f)$*

$$
\begin{aligned}
&\mathcal{L}_{\mathrm{NCE}}(f) - \sqrt{\mathrm{Var}(f(x) \mid y)} - \frac{1}{2}\mathrm{Var}(f(x) \mid y) - \mathcal{O}\left(M^{-1/2}\right) \\
&\leq \mathcal{L}_{\mathrm{CE}}^{\mu}(f) + \log(M/K) \leq \mathcal{L}_{\mathrm{NCE}}(f) + \sqrt{\mathrm{Var}(f(x)|y)} + \mathcal{O}\left(M^{-1/2}\right),
\end{aligned}
\tag{3}
$$

*where $\log(M/K)$ is a constant\*, $\mathrm{Var}(f(x)|y) = \mathbb{E}_{p(y)}\left[\mathbb{E}_{p(x|y)}\|f(x) - \mathbb{E}_{p(x|y)}f(x)\|^2\right]$ denotes the conditional (intra-class) feature variance, and $\mathcal{O}\left(M^{-1/2}\right)$ denotes the order of the approximation error by using $M$ negative samples.*

Notably, our generalization bounds above improve over previous ones in the following aspects:

1) we do not require the conditional independence assumption as in Saunshi et al. (2019);

2) we directly analyze the widely adopted InfoNCE loss (for contrastive learning) and CE loss (for supervised finetuning), while Saunshi et al. (2019) are restricted to hinge and logistic objectives that have worse performance in practice (Chen et al., 2020a);

3) the class collision error terms introduced in Saunshi et al. (2019) (due to the existence of same-class samples in the negative samples) now disappear in our bounds by adopting the InfoNCE loss, which also helps understand why InfoNCE performs better in practice; and

4) the bounds in Saunshi et al. (2019) will become looser with more negative samples, which is contradictory to the common practice (Chen et al., 2020a). While in our bounds, a larger $M$ indeed has a lower approximation error and helps close the generalization gap.

In fact, several recent works have also been devoted to resolve the last "large-$M$" problem (Ash et al., 2021; Merad et al., 2020). Nevertheless, their analysis also requires the conditional independence assumption as in Saunshi et al. (2019), while we show this problem can be resolved even without conditional independence. Nozawa & Sato (2021) also establish bounds for the InfoNCE loss, but their bounds have incompressible class collision terms while ours do not.

Nevertheless, an important message of the theorem above is that Assumption 4.1 alone is still **insufficient** to guarantee good downstream performance. As there are intra-class variance terms in the upper and lower bounds, when they are large enough, contrastive learning might still have inferior performance as shown in Proposition 3.1. Although the variance terms can be easily eliminated with the canonical conditional independence assumption, discussions in Section 1 have already demonstrated its impracticality. In the next part, we will present a new understanding of how contrastive learning could control this variance term in practice.

## 4.2 CLOSING THE GAP WITH INTRA-CLASS CONNECTIVITY

The theorem above motivates us to study how contrastive learning could effectively control its intra-class variance and learn class-separated features. Here, we propose a new understanding of this clustering ability through a dissection of the augmented views. In particular, we notice that although samples are different from each other, applying aggressive augmentations like that in SimCLR (Chen

---

\*$\log(M/K)$ could be absorbed in the loss functions by replacing sum with mean in InfoNCE and CE.

et al., 2020a) can largely make them more alike. For example, in Figure 1(b), two different cars become very similar when they are both cropped to the wheels. Then, with contrastive learning, the two cars will have closer representations as they share a common view of the wheels. In other words, two different intra-class samples could be aligned together if they have overlapped augmented views. If all intra-class samples could be bridged by data augmentations, we can successfully cluster the whole class together. Below, we formalize the intuition above with the language of graphs.

**Notations.** A graph $\mathcal{G}$ is represented by a tuple $\mathcal{G} = (\mathcal{V}, \mathcal{E})$ where $\mathcal{V} = (v_1, v_2, \ldots, v_N)$ is a set of vertices and $\mathcal{E} \subseteq \mathcal{V} \times \mathcal{V}$ is a set of edges. A path is a sequence of edges that joins a sequence of vertices, *e.g.*, $v_{i_1} - v_{i_2} - \cdots - v_{i_k}$. We say that two vertices $v$ and $u$ are connected if $\mathcal{G}$ contains a path from $v$ to $u$. A graph is said to be connected if every pair of vertices in the graph is connected. Two graphs are said to be disjoint if any pair of inter-graph vertices are not connected.

To begin with, we define the concept of $\mathcal{T}$-connectivity of sample pairs, which describes whether two samples could be connected via the augmentation overlap of their augmented views.

**Definition 4.3** ($\mathcal{T}$-connectivity). *Given a collection of augmentations $\mathcal{T} = \{t \mid t : \mathbb{R}^d \to \mathbb{R}^d\}$, we say that two different images $x_i, x_j \in \mathbb{R}^d$ are $\mathcal{T}$-connected if they have overlapped views:* $\mathrm{supp}(p(x_i^+|x_i)) \bigcap \mathrm{supp}(p(x_j^+|x_j)) \neq \varnothing$, *or equivalently,* $\exists\, t_i, t_j \in \mathcal{T}$ *such that* $t_i(x_i) = t_j(x_j)$.

Then, we can define an augmentation graph of all training samples in terms of their $\mathcal{T}$-connectivity.

**Definition 4.4** (Augmentation Graph). *Given a set of $N$ samples $\mathcal{D} = \{x_i\}_{i=1}^N$ and an augmentation set $\mathcal{T} = \{t \mid t : \mathbb{R}^d \to \mathbb{R}^d\}$, we can define an augmentation graph $\mathcal{G}(\mathcal{D}, \mathcal{T}) = (\mathcal{V}, \mathcal{E})$ as*

- *we take the $N$ natural samples as the vertices of the graph, i.e., $\mathcal{V} = \{x_i\}_{i=1}^N$;*

- *there exists an edge $e_{ij}$ between two vertices $x_i$ and $x_j$ if they are $\mathcal{T}$-connected.*

Based on these concepts, we introduce the following assumption that with a proper choice of data augmentations, all intra-class samples could form a connected graph, as depicted in Figure 3(a).

**Assumption 4.5** (Intra-class Connectivity). *Given a training set $\mathcal{D}_u$, there exists an appropriate augmentation set $\mathcal{T}$ such that the augmentation graph $\mathcal{G}(\mathcal{D}_u, \mathcal{T})$ is class-wise connected, i.e., $\forall\, k \in \{1, \ldots, K\}$, the subgraph $\mathcal{G}_k$ (graph $\mathcal{G}$ restricted to vertices in class $k$) is connected.*

Comparing to Saunshi et al. (2019) and Lee et al. (2020) that require (nearly) conditional independence $p(x, x^+|y) = p(x|y)p(x^+|y)$, ours only requires the connectivity of intra-class samples as in Figure 1(b), and does not need them to be conditionally independent.

To make this analysis technically simpler, we make another assumption that we can align positive samples perfectly by minimizing the InfoNCE loss. In practice, the alignment loss can typically be minimized up to a small error $\varepsilon$, and we have appended a more involved discussion of this weak alignment scenario in Appendix B. For now, we focus on the simplified perfect alignment scenario.

**Assumption 4.6** (Perfect Alignment). *At the minimizer $f^\star$ of the InfoNCE loss, we can achieve perfect alignment, i.e., $\forall\, x, x^+ \sim p(x, x^+), f^\star(x) = f^\star(x^+)$.*

**Proposition 4.7.** *Under Assumptions 4.5 & 4.6, by minimizing the InfoNCE loss we can conclude that the conditional variance terms vanish at the minimizer $f^\star$, i.e.,*

$$\mathrm{Var}(f^\star(x) \mid y) = 0. \tag{4}$$

Intuitively, for samples in each class $k$, if the corresponding subgraph $\mathcal{G}_k$ is connected, there exists a path connecting every intra-class pairs $(x_i, x_j)$, as shown in Figure 3(a). Consequently, aligning the positive pairs will also align all samples on the path, and eventually align $x_i$ and $x_j$. In this way, all intra-class samples can be clustered together and the intra-class variance shrinks to zero (under Assumption 4.6). Besides, because proper data augmentation will not cause inter-class augmentation overlap (Assumption 4.1), inter-class samples can be well separated with the uniformity term. As a result, we can attain alignment of intra-class samples while maximizing the uniformity of inter-class samples. According to Theorem 4.2, we will have an asymptotically closed generalization gap (with more negative samples $M \to \infty$) for the encoder that minimizes the contrastive loss.

**Theorem 4.8** (Guarantees for the Optimal Encoder). *If Assumption 4.1, 4.5 & 4.6 hold and $f$ is $L$-smooth, then, for the minimizer $f^\star = \arg\min \mathcal{L}_{\mathrm{NCE}}(f)$, its classification risk can be upper and lower bounded by its contrastive risk as*

$$\mathcal{L}_{\mathrm{NCE}}(f^\star) - \mathcal{O}\left(M^{-1/2}\right) \leq \quad \mathcal{L}_{\mathrm{CE}}^\mu(f^\star) + \log(M/K) \leq \mathcal{L}_{\mathrm{NCE}}(f^\star) + \mathcal{O}\left(M^{-1/2}\right). \tag{5}$$

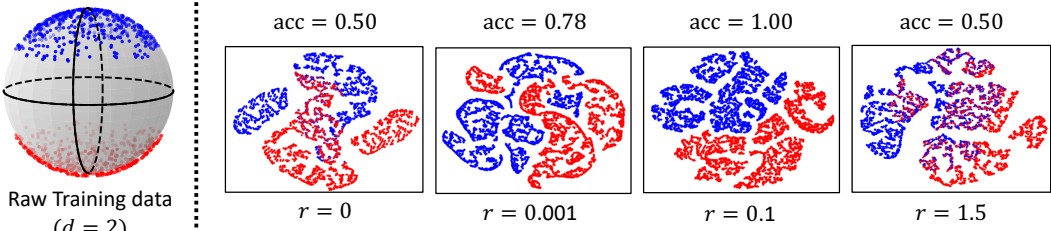

Figure 4: t-SNE visualization of features learned with different augmentation strength $r$ on the random augmentation graph experiment. Each dot denotes a sample and its color denotes its class.

We note that different to previous bounds that hold for any $f \in \mathcal{F}$ as in Theorem 4.2, our results here only stand for the minimizer of the contrastive loss $\hat{f}^\star$. This indicates that the InfoNCE loss alone cannot simply guarantee good downstream performance, and the learning dynamics matters for the contrastive learning to learn useful features.

### 4.3 Rethinking the Role of Data Augmentations

Our analysis above suggests a new understanding of the role of data augmentations in contrastive learning. Conventionally, the success of contrastive learning is usually attributed to learning invariance *w.r.t.* various data augmentations by matching positive examples. However, as shown in Proposition 3.1, matching positive pairs alone is theoretically inadequate to learn useful features. Indeed, assuming that an ideal encoder that possesses invariance *a priori* does exist, like invariance to translation (CNNs), rotation (Cheng et al., 2016), and scaling (Xu et al., 2014), do we obtain class-discriminative features simply by random initialization? Still NO, since these low-level properties are independent of high-level class information that we want to learn. Thus, the reason why contrastive learning works cannot simply be attributed to the invariance learning principle.

We instead believe that the role of data augmentation is to create a certain degree of "chaos" between different intra-class samples (Figure 1(b)) such that they become more alike (or formally, $\mathcal{T}$-connected). In this way, the chaos serves as a "ladder" for bridging intra-class samples together when labels are absent, and the mission of the contrastive loss is to "climb this ladder", that is, aligning intra-class samples by aligning the overlapped positive samples, as shown in Figure 3(a). Therefore, from our perspective, instance discrimination by contrastive learning is actually **a surrogate** for the classification task, and the surrogate can complete its misson when the ladder of chaos is complete (or formally, when intra-class connectivity holds).

## 5 Quantifying the Influence of Augmentation Strength

We have shown that with appropriate augmentations, we can derive guarantees on downstream performance. However, in practice, as illustrated in Figure 3(b), there could be cases where augmentations are either too weak (intra-class features cannot be clustered together as in Figure 2) or too strong (inter-class features will also collapse to the same point) and lead to sub-optimal results. In this section, we further provide a quantitative analysis of how different strength of data augmentation will affect the final performance, both theoretically and empirically.

### 5.1 Characterization on Random Augmentation Graph

In practice, there are various data augmentation types that are hard to be described precisely. For the ease of analysis, we consider a simple case where for each class $k$, there are $N$ samples uniformly distributed around the cluster center $c_k$ on a hypersphere $\mathbb{S}^d$. We then augment each sample $x_i$ with random samples in a hyper-disk of radius $r$ on the hypersphere.

In Appendix D, we provide theoretical analysis on how different augmentation strength (measured by $r$) will affect the connectivity of the augmentation as a function of the number of samples $N$, the position of the cluster centers $c_k$ and input dimensions $d$. In particular, the minimal $r$ for the graph to be connected decreases as $N$ increases, so large-scale datasets can bring better connectivity. Meanwhile, the required $r$ also increases as $d$ increases, so we need more samples or stronger augmentations for large-size inputs. Here, we show our simulation results by applying contrastive

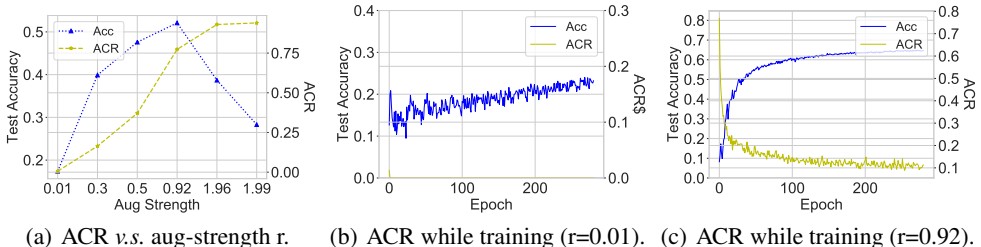

(a) ACR *v.s.* aug-strength r.   (b) ACR while training (r=0.01).   (c) ACR while training (r=0.92).

Figure 5: (a) Average Confusion Rate (ACR) and downstream accuracy *v.s.* different augmentation strength (before training). (b,c): ACR and downstream accuracy while training.

learning to the problem above. From Figure 4, we can see that when $r = 0$ (no augmentation), the features are mixed together and hardly (linearly) separable, which corresponds to the under-overlap case in Figure 3(b). As we increase $r$ from 0 to 0.1, the features become more and more discriminative. And when $r$ is too large ($r = 1.5$), the inter-class features become mixed and inseparable again (over-overlap). In Appendix C.2, we provide visualization results of the augmentation graphs, which also align well with our analysis. Overall, our theoretical and empirical discussions verify our theory that intra-class augmentation overlap with a proper amount of data augmentation is crucial for contrastive learning to work well.

## 5.2 NEW SURROGATE METRICS FOR AUGMENTATION OVERLAP

From our theory and analysis above, we see that the augmentation overlap between intra-class samples indeed matters from contrastive learning to generalize better. Inspired by this, we propose the Confusion Ratio metric as a measure of the degree of augmentation overlap. Specifically, for an unlabeled dataset $\mathcal{D}_u$ with $N$ samples, we randomly augment each raw sample $x_i \in \mathcal{D}_u$ for $C$ times, and get an augmented set $\widetilde{\mathcal{D}}_u = \{x_{ij}, i \in [N], j \in [C]\}$. Then, for each $x_{ip} \in \widetilde{\mathcal{D}}_u$ that is an augmented view of $x_i \in \mathcal{D}_u$, denoting its $k$-nearest neighbors in $\widetilde{\mathcal{D}}_u$ in the feature space of $f$ as $\mathcal{N}_k(x_{ip}, f)$ and other augmented views from the same image as $\mathcal{C}(x_{ip}) = \{x_{ij}, j \neq p\}$, we can define its Confusion Ratio (CR) as the ratio of augmented views from **different** raw samples in its $k$-nearest neighbors,

$$\text{CR}(x_{ij}, f) = \frac{\#[\mathcal{N}_k(x_{ip}, f) \setminus \mathcal{C}(x_{ip})]}{\#\mathcal{N}_k(x_{ip}, f)} \in [0, 1]. \tag{6}$$

We also define its average as Average Confusion Ratio (ACR):

$$\text{ACR}(f) = \mathbb{E}_{x_{ij} \sim \widetilde{\mathcal{D}}_u} \text{CR}(x_{ij, f}). \tag{7}$$

When augmentation overlap happens, the nearest neighbors could be augmented views from a different sample, leading to a higher ACR. Thus, ACR measures the degree of augmentation overlap, and a higher ACR indicates a higher degree of augmentation overlap. Here we take $k = 1$ by default.

Here, to measure the augmentation strength in real-world datasets, following the common practice (Chen et al., 2020a), we adopt the $\text{RandomResizedCrop}$ operator with scale range $[a, b]$ for data augmentation, and we define its strength of augmentation as $r = (1 - b) + (1 - a)$ (a comparison with other kinds of augmentations, *e.g.,* color jittering, can be found in Appendix C.1). As shown in Figure 5(a), ACR (augmentation overlap) indeed increases with the strength of data augmentations, and only a moderate ACR achieves the best accuracy, which is consistent with our theory discussed above. Besides, we also plot the change of ACR along the training process in Figure 5(b) & 5(c). We can notice that for weak augmentations, the initial ACR is low, and it rapidly decreases to zero and seldom changes while training, which leads to poor test accuracy. Instead, with proper augmentations, the initial ACR is higher, and it gradually decreases to zero and obtains good accuracy. This is also consistent with our theory that we need a certain amount of augmentation overlap for contrastive learning to work well. At the beginning, this will lead to a higher ACR, but as training continues, better alignment (lower ACR) will help bring up the test accuracy.

**Average Relative Confusion (ARC).** In the discussion above, we notice that ACR itself does not indicate the test accuracy, but the relative change of ACR before and after training can be used as such an indicator. A large change of ACR means a large change of augmentation overlap, which indicates that the contrastive loss can actually cluster intra-class samples together through overlapped

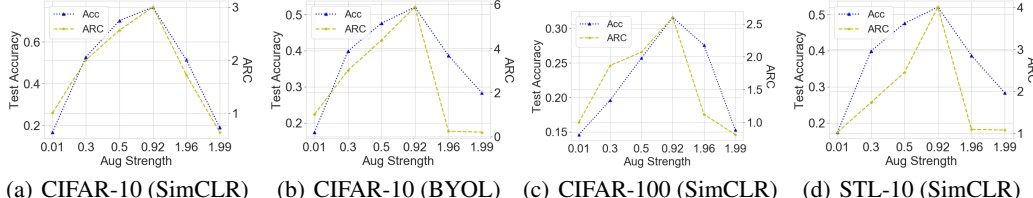

(a) CIFAR-10 (SimCLR)  (b) CIFAR-10 (BYOL)  (c) CIFAR-100 (SimCLR)  (d) STL-10 (SimCLR)

Figure 6: Average Relative Confusion (ARC) and downstream accuracy *v.s.* different augmentation strength on different datasets (CIFAR-10, CIFAR-100, and STL-10) with different contrastive learning methods: SimCLR (Chen et al., 2020a) and BYOL (Grill et al., 2020).

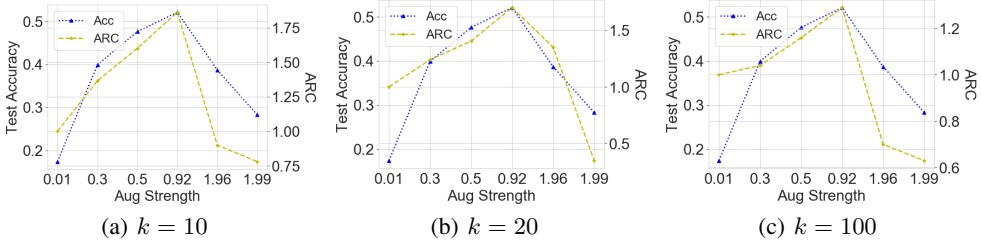

(a) $k = 10$       (b) $k = 20$       (c) $k = 100$

Figure 7: Average Relative Confusion (ARC) and downstream accuracy *v.s.* different augmentation strength on CIFAR-10 (SimCLR) with different number of nearest neighbors $k$.

views. Based on this observation, we propose Average Relative Confusion (ARC) as

$$\text{ARC} = \frac{1 - \text{ACR}(f_{\text{final}})}{1 - \text{ACR}(f_{\text{init}})}, \tag{8}$$

a ratio calculated with the initial ACR of the initialized model $f_{\text{init}}$ and the final ACR of the pre-trained model $f_{\text{final}}$. A higher ARC indicates that the contrastive learning process faces a hard task (augmentation overlap) at the beginning (high initial ACR), while successfully clustering intra-class samples with good alignment of positive samples at the end (lo final ACR). Therefore, a higher ARC score should correspond to higher downstream accuracy.

As shown in Figure 6 & 7, as augmentations become stronger, ARC scores indeed align well with the change of downstream accuracy across 1) different datasets, 2) different contrastive methods, and 3) different choices of $k$. This justifies our understanding of contrastive learning through augmentation overlap. Meanwhile, as the calculation of ARC only involves unsupervised data, it could serve as a good surrogate metric for evaluating contrastive learning without using labeled data. Compared to previous evaluation methods like linear classification (Eq. 2), our ARC metric is more preferable as 1) it is theoretically motivated; 2) it does not need labeled data; 3) it does not need to learn additional modules like linear classifiers or rotation tasks (Reed et al., 2021). More experimental details can be found in Appendix E.

## 6 CONCLUSION

In this paper, we have proposed a new understanding of contrastive learning through a revisiting of the role of data augmentations. In particular, we notice the aggressive data augmentation applied in contrastive learning can significantly increase the augmentation overlap between intra-class samples, and as a result, by aligning positive samples, we can also cluster inter-class samples together. Based on this insight, we develop a new augmentation overlap theory that could guarantee good downstream performance without relying on conditional independence and obtain asymptotically closed gaps. With this perspective, we also characterize how different augmentation strength affects downstream performance with both random graphs and real-world datasets. Last but not least, we also develop a new surrogate metric for evaluating contrastive learning without labels and show that it aligns well with downstream performance. Overall, we believe that we pave a new way for understanding contrastive learning with insights on the designing of contrastive methods and evaluation metrics.

## ACKNOWLEDGEMENT

Yisen Wang is partially supported by the National Natural Science Foundation of China under Grant 62006153, Project 2020BD006 supported by PKU-Baidu Fund, and Huawei Technologies Inc. Jiansheng Yang is supported by the National Science Foundation of China under Grant No. 11961141007. Zhouchen Lin is supported by the NSF China (No. 61731018), NSFC Tianyuan Fund for Mathematics (No. 12026606), Project 2020BD006 supported by PKU-Baidu Fund, and Qualcomm.

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

# A  OMITTED PROOFS

## A.1  PROOF OF PROPOSITION 3.1

**Proposition A.1** (Class-uniform features also minimize the InfoNCE loss). *For $N$ training examples of $K$ classes, consider the case when features $\{f(x_i)\}_{i=1}^N$ are randomly distributed in $\mathbb{S}^{m-1}$ with maximal uniformity while also satisfying $\forall x_i, x_i^+ \sim p(x, x^+), f(x_i) = f(x_i^+)$. Because we have perfect alignment and perfect uniformity, the InfoNCE loss achieves its minimum. However, the downstream classification accuracy is at most $1/K + \varepsilon$ and $\varepsilon$ is nearly zero when $N$ is large enough.*

*Proof.* We only need to give a counterexample that satisfy the desired classification accuracy. We consider the case when there is no $\mathcal{T}$-connectivity between any pair of samples from $\{x_i\}_{i=1}^N$, which is easily achieved if we adopt a small enough data augmentation. In this scenario, the perfect alignment of positive samples $(x_i, x_i^+)$ could have no effect on the other samples. Therefore, when the features $\{f(x_i)\}_{i=1}^N$ are uniformly distributed in $\mathbb{S}^{m-1}$, according to the law of large number, for any measurable set $\mathcal{U} \in \mathbb{S}^{m-1}$, when $N$ is large enough, there will be almost equal size of features from each class in $\mathcal{U}$. Consequently, any classifier $g$ that classifies $\mathcal{U}$ to class $k$ will only have $1/K$ accuracy asymptotically. $\qquad\square$

## A.2  PROOF OF THEOREM 4.2

We will prove the upper and lower bounds separately as follows.

### A.2.1  THE UPPER BOUND

We first provide the upper bound of the approximation error of the following Monte Carlo estimate.

**Lemma A.2.** *For $\text{LSE} := \log \mathbb{E}_{p(z)} \exp(f(x)^\top g(z))$, we denote its (biased) Monte Carlo estimate with $M$ random samples $z_i \sim p(z), i = 1, \ldots, M$ as $\widehat{\text{LSE}}_M = \log \frac{1}{M} \sum_{i=1}^M \exp(f(x)^\top g(z_i))$. Then the approximation error $A(M)$ can be upper bounded in expectation as*

$$A(M) := \mathbb{E}_{p(x, z_i)} |\widehat{\text{LSE}}(M) - \text{LSE}| \leq \mathcal{O}(M^{-1/2}). \tag{9}$$

*We can see that the approximation error converges to zero in the order of $1/M^{-1/2}$.*

*Proof.* First, we have

$$\mathbb{E}_{p(x, z_i)} \left[ \log \frac{1}{M} \sum_{i=1}^M \exp(f(x)^\top g(z_i)) - \log \mathbb{E}_{p(z_i)} \exp(f(x)^\top g(z_i)) \right]$$

$$\leq e \mathbb{E}_{p(x, z_i)} \left[ \frac{1}{M} \sum_{i=1}^M \exp(f(x)^\top g(z_i)) - \mathbb{E}_{p(z_i)} \exp(f(x)^\top g(z_i)) \right] = \mathcal{O}(M^{-1/2}),$$

where the first inequality follows the Intermediate Value Theorem and $e$ (the natural number) is the upper bound of the absolute derivative of log between two points when $|f(x)^\top g(z_i)| \leq 1$. And the second inequality follows the Berry-Esseen Theorem given the bounded support of $\exp(f(x)^\top g(z_i))$ as following: for i.i.d random variables $Y_i$ with bounded support $\text{supp}(Y) \subset$

$[-\alpha, \alpha]$, zero mean and bounded variance $\sigma_Y^2 < \alpha^2$, we have:

$$
\mathbb{E}\left[\left\|\frac{1}{M}\sum_{i=1}^{M}Y_i\right\|\right] = \frac{\sigma_y}{\sqrt{M}}\mathbb{E}\left[\left\|\frac{1}{\sqrt{M}\sigma_y}\sum_{i=1}^{M}Y_i\right\|\right]
$$
$$
=\frac{\sigma_Y}{\sqrt{M}}\int_0^{\frac{\alpha\sqrt{M}}{\sigma_Y}}\mathbb{P}\left[\left\|\frac{1}{\sqrt{M}\sigma_Y}\sum_{i=1}^{M}Y_i\right\| > x\right]\mathrm{d}x
$$
$$
\leq\frac{\sigma_Y}{\sqrt{M}}\int_0^{\frac{\alpha\sqrt{M}}{\sigma_Y}}\mathbb{P}[|\mathcal{N}(0,1)| > x] + \frac{C_\alpha}{\sqrt{M}}\mathrm{d}x
$$
$$
\leq\frac{\sigma_Y}{\sqrt{M}}\left(\frac{\alpha C_\alpha}{\sigma_Y} + \int_0^\infty\mathbb{P}[|\mathcal{N}(0,1)| > x]\mathrm{d}x\right)
$$
$$
\leq\frac{C_\alpha}{\sqrt{M}} + \frac{\alpha}{\sqrt{M}}\mathbb{E}[|\mathcal{N}(0,1)|] = \mathcal{O}(M^{-1/2})
$$

where the constant $C_\alpha$ only depends on $\alpha$. Here, we set $Y_i = \exp(f(x)^\top g(z_i)) - \mathbb{E}_{p(z_i)}\exp(f(x)^\top g(z_i))$. As $|f(x)^\top g(z_i)| \leq 1$, $|Y_i| \leq 2e$. $Y_i$ has zero mean and bounded variance $(2e)^2$. $\qquad\square$

**Theorem A.3.** *For each $f \in \mathcal{F}$, the mean CE loss can be upper bounded by the InfoNCE loss:*

$$
\mathcal{L}_{\mathrm{CE}}^\mu(x,y;f) \leq \mathcal{L}_{\mathrm{NCE}}(x;f) - \log(M/K) + \sqrt{\mathrm{Var}(f(x)\mid y)} + A(M), \tag{10}
$$

*where $\mathrm{Var}(f(x)|y) = \mathbb{E}_{p(y)}\left[\mathbb{E}_{p(x|y)}\|f(x) - \mathbb{E}_{p(x|y)}f(x)\|^2\right]$ denotes the conditional variance.*

*Proof.* Denote $p(x, x^+, y)$ as the joint distribution of the positive pairs $x, x^+$ and the label $y$. Denote the $M$ independently negative smaples as $\{x_i^-\}_{i=1}^M$. According to Assumption 4.1, $x^+$ and $x$ here has the same label $y$. Denote $\mu_y$ as the center of features of class $y$, $y = 1, \ldots, K$. Then we have the following lower bounds of the InfoNCE loss,

$$
\mathcal{L}_{\mathrm{NCE}}(f) = -\mathbb{E}_{p(x,x^+)}f(x)^\top f(x^+) + \mathbb{E}_{p(x)}\mathbb{E}_{p(x_i^-)}\log\sum_{i=1}^{M}\exp(f(x)^\top f(x_i^-))
$$

$$
= -\mathbb{E}_{p(x,x^+)}f(x)^\top f(x^+) + \mathbb{E}_{p(x)}\mathbb{E}_{p(x_i^-)}\log\frac{1}{M}\sum_{i=1}^{M}\exp(f(x)^\top f(x_i^-)) + \log M
$$

$$
\overset{(1)}{\geq} -\mathbb{E}_{p(x,x^+)}f(x)^\top f(x^+) + \mathbb{E}_{p(x)}\log\frac{1}{M}\mathbb{E}_{p(x_i^-)}\sum_{i=1}^{M}\exp(f(x)^\top f(x_i^-)) - A(M) + \log M
$$

$$
= -\mathbb{E}_{p(x,x^+)}f(x)^\top f(x^+) + \mathbb{E}_{p(x)}\log\mathbb{E}_{p(x^-)}\exp(f(x)^\top f(x^-)) - A(M) + \log M
$$

$$
= -\mathbb{E}_{p(x,x^+,y)}f(x)^\top f(x^+) + \mathbb{E}_{p(x)}\log\mathbb{E}_{p(y^-)}\mathbb{E}_{p(x^-|y^-)}\exp(f(x)^\top f(x^-)) - A(M) + \log M
$$

$$
\overset{(2)}{\geq} -\mathbb{E}_{p(x,x^+,y)}f(x)^\top f(x^+) + \mathbb{E}_{p(x)}\log\mathbb{E}_{p(y^-)}\exp(\mathbb{E}_{p(x^-|y^-)}\left[f(x)^\top f(x^-)\right]) - A(M) + \log M
$$

$$
= -\mathbb{E}_{p(x,x^+,y)}f(x)^\top(\mu_y + f(x^+) - \mu_y) + \mathbb{E}_{p(x)}\log\mathbb{E}_{p(y^-)}\exp(\mathbb{E}_{p(x^-|y^-)}\left[f(x)^\top f(x^-)\right]) - A(M) + \log M
$$

$$
= -\mathbb{E}_{p(x,x^+,y)}[f(x)^\top\mu_y + f(x)^\top(f(x^+) - \mu_y)] + \mathbb{E}_{p(x)}\log\mathbb{E}_{p(y^-)}\exp(f(x)^\top\mu_{y^-}) - A(M) + \log M
$$

$$
\overset{(3)}{\geq} -\mathbb{E}_{p(x,x^+,y)}\left[f(x)^\top\mu_y + \|(f(x^+) - \mu_y)\|\right] + \mathbb{E}_{p(x)}\log\mathbb{E}_{p(y^-)}\exp(f(x)^\top\mu_{y^-}) - A(M) + \log M
$$

$$
\overset{(4)}{\geq} -\mathbb{E}_{p(x,y)}f(x)^\top\mu_y - \sqrt{\mathbb{E}_{p(x,y)}\|f(x) - \mu_y\|^2} + \mathbb{E}_{p(x)}\log\mathbb{E}_{p(y^-)}\exp(f(x)^\top\mu_{y^-}) - A(M) + \log M
$$

$$
= -\mathbb{E}_{p(x,y)}f(x)^\top\mu_y - \sqrt{\mathrm{Var}(f(x)\mid y)} + \mathbb{E}_{p(x)}\log\frac{1}{K}\sum_{k=1}^{K}\exp(f(x)^\top\mu_k) - A(M) + \log M
$$

$$
= \mathbb{E}_{p(x,y)}\left[-f(x)^\top\mu_y + \log\sum_{k=1}^{K}\exp(f(x)^\top\mu_k)\right] - \sqrt{\mathrm{Var}(f(x)\mid y)} - A(M) + \log(M/K)
$$

$$
= \mathcal{L}_{\mathrm{CE}}^\mu(f) - \sqrt{\mathrm{Var}(f(x)\mid y)} - A(M) + \log(M/K),
$$

which is equivalent to our desired results. In the proof above, (1) follows Lemma A.2; (2) follows the Jensen's inequality for the convex function $\exp(\cdot)$; (3) follows from the fact that because $f(x) \in \mathbb{S}^{m-1}$, we have

$$f(x)^\top (f(x^+) - \mu_y) \leq \left( \frac{f(x^+) - \mu_y}{\|f(x^+) - \mu_y\|} \right)^\top (f(x^+) - \mu_y) = \|f(x^+) - \mu_y\|; \qquad (11)$$

and (4) follows the Cauchy–Schwarz inequality and the fact that because $p(x, x^+) = p(x^+, x)$ holds, $x, x^+$ have the same marginal distribution. $\qquad \square$

### A.2.2 THE LOWER BOUND

In this part, we further show a lower bound on the downstream performance.

**Lemma A.4** (Budimir et al. (2000) Corollary 3.5 (restated)). *Let $g : \mathbb{R}^m \to \mathbb{R}$ be a differentiable convex mapping and $z \in \mathbb{R}^n$. Suppose that $g$ is L- smooth with the constant $L > 0$, i.e., $\forall x, y \in \mathbb{R}^m$, $\|\nabla g(x) - \nabla g(y)\| \leq L\|x - y\|$. Then we have*

$$0 \leq \mathbb{E}_{p(z)} g(z) - g\left(\mathbb{E}_{p(z)} z\right) \leq L \left[ \mathbb{E}_{p(z)}\|z\|^2 - \|\mathbb{E}_{p(z)} z\|^2 \right] = L \sum_{j=1}^{n} \mathrm{Var}(z^{(j)}), \qquad (12)$$

*where $x^{(j)}$ denotes the j-th dimension of $x$.*

With the lemma above, we can derive the lower bound of the downstream performance.

**Theorem A.5.** *For any $f \in \mathcal{F}$, we have*

$$L_{\mathrm{CE}}^\mu(f) \geq \mathcal{L}_{\mathrm{NCE}}(x; f) - \sqrt{\mathrm{Var}(f(x) \mid y)} - \frac{1}{2} \mathrm{Var}(f(x) \mid y) - A(M) - \log \frac{M}{K}, \qquad (13)$$

*where $\mathrm{Var}(u(x)|y) = \mathbb{E}_{p(y)} \left[ \mathbb{E}_{p(x|y)} \|u(x) - \mathbb{E}_{p(x|y)} u(x)\|^2 \right]$ denotes the conditional variance.*

*Proof.* Similar to the proof of Theorem A.3, we have

$$L_{\mathrm{CE}}^\mu(f) = -\mathbb{E}_{p(x,y)} f(x)^\top \mu_y + \mathbb{E}_{p(x)} \log \sum_{i=1}^{K} \exp(f(x)^\top \mu_i)$$

$$= -\mathbb{E}_{p(x,y)} f(x)^\top \mu_y + \mathbb{E}_{p(x)} \log \frac{1}{K} \sum_{i=1}^{K} \exp(f(x)^\top \mu_i) + \log K$$

$$= -\mathbb{E}_{p(x,y)} f(x)^\top \mu_y + \mathbb{E}_{p(x)} \log \mathbb{E}_{p(y_i^-)} \exp(f(x)^\top \mu_{y_i}) + \log K$$

$$\overset{(1)}{\geq} -\mathbb{E}_{p(x,y)} [f(x)^\top f(x^+) + f(x)^\top (\mu_y - f(x^+))] + \mathbb{E}_{p(x)} \mathbb{E}_{p(y_i^-)} \log \frac{1}{M} \sum_{i=1}^{M} \exp(f(x)^\top \mu_{y_i}) - A(M) + \log K$$

$$\overset{(2)}{\geq} -\mathbb{E}_{p(x,x^+)} f(x)^\top f(x^+) - \mathbb{E}_{p(x,y)} \|f(x)^\top - \mu_y\|$$

$$+ \mathbb{E}_{p(x)} \mathbb{E}_{p(y_i^-)} \log \frac{1}{M} \sum_{i=1}^{M} \exp(\mathbb{E}_{p(x_i^- | y_i^-)} f(x)^\top f(x_i^-)) - A(M) + \log K$$

$$\overset{(3)}{\geq} -\mathbb{E}_{p(x,x^+)} f(x)^\top f(x^+) - \sqrt{\mathrm{Var}(f(x) \mid y)}$$

$$+ \mathbb{E}_{p(x)} \mathbb{E}_{p(y_i^-)} \mathbb{E}_{p(x_i^- | y)} \left[ \log \frac{1}{M} \sum_{i=1}^{M} \exp(f(x)^\top f(x^-)) \right] - \frac{1}{2} \sum_{j=1}^{m} \mathrm{Var}(f_j(x^-) \mid y) - A(M) + \log K$$

$$= -\mathbb{E}_{p(x,x^+)} \left[ f(x)^\top f(x^+) + \mathbb{E}_{p(x_i^-)} \log \sum_{i=1}^{M} \exp(f(x)^\top f(x^-)) \right]$$

$$- \sqrt{\mathrm{Var}(f(x) \mid y)} - \frac{1}{2} \sum_{j=1}^{m} \mathrm{Var}(f_j(x^-) \mid y) - A(M) + \log K - \log M$$

$$\overset{(4)}{\geq} \mathcal{L}_{\mathrm{NCE}}(x; f) - \sqrt{\mathrm{Var}(f(x) \mid y)} - \frac{1}{2} \mathrm{Var}(f(x) \mid y) - A(M) - \log \frac{M}{K},$$

which is our desired result. In the proof, (1) we adopt a Monte Carlo estimate with $M$ samples from $p(y)$ and bound the approximation error with Lemma A.2; (2) follows the same deduction in Theorem A.3; (3) the first term is derived following the Cauchy–Schwarz inequality for the alignment term. As for the second term, we first show that the convex function logsumexp is $L$-smooth as a function of $f(x_j^-)$ in our scenario. Because $\|f(X)\| \leq 1$, we have $\forall f(x_{j_1}), f(x_{j_2}) \in \mathbb{R}^m$, the following bound on the difference of their gradients holds

$$
\begin{aligned}
&\left\| \frac{\partial \log[\exp(f(x)^\top f(x_{j_1}^-) + \sum_{i \neq j} \exp(f(x)^\top f(x_i^-)))]}{\partial f(x_{j_1}^-)} - \frac{\partial \log[\exp(f(x)^\top f(x_{j_2}^-) + \sum_{i \neq j} \exp(f(x)^\top f(x_i^-)))]}{\partial f(x_{j_2}^-)} \right\| \\
&= \left\| \left( \frac{\exp(f(x)^\top f(x_{j_1}^-))}{\exp(f(x)^\top f(x_{j_1}^-) + \sum_{i \neq j} \exp(f(x)^\top f(x_i^-)))} - \frac{\exp(f(x)^\top f(x_{j_2}^-))}{\exp(f(x)^\top f(x_{j_2}^-) + \sum_{i \neq j} \exp(f(x)^\top f(x_i^-)))} \right) f(x) \right\| \\
&\leq \left| \frac{(\sum_{i \neq j} \exp(f(x)^\top f(x_i^-))) \exp(f(x)^\top f(x_{j_1}^-)) - \sum_{i \neq j} \exp(f(x)^\top f(x_i^-)) \exp(f(x)^\top f(x_{j_2}^-))}{(\exp(f(x)^\top f(x_{j_1}^-)) + \sum_{i \neq j} \exp(f(x)^\top f(x_i^-)))(\exp(f(x)^\top f(x_{j_2}^-)) + \sum_{i \neq j} \exp(f(x)^\top f(x_i^-)))} \right| \cdot \|f(x)\| \\
&\leq \|f(x)\| \leq \frac{1}{2} \left\| f(x_{j_1}^-) - f(x_{j_2}^-) \right\|
\end{aligned}
$$

So here the logsumexp is $L$-smooth for $L = \frac{1}{2}$. Then, we can apply the reversed Jensen's inequality in Lemma A.4; (4) holds because

$$
\begin{aligned}
&\sum_{j=1}^m \mathrm{Var}(f_j(x)|y) \\
&= \sum_{j=1}^m \mathbb{E}_{p(y)} \mathbb{E}_{p(x|y)} (f_j(x) - \mathbb{E}_{p(x'|y)} f_j(x'))^2 \\
&= \mathbb{E}_{p(y)} \mathbb{E}_{p(x|y)} \sum_{j=1}^m (f_j(x) - \mathbb{E}_{x'} f_j(x'))^2 \\
&= \mathbb{E}_{p(y)} \mathbb{E}_{p(x|y)} \| f(x) - \mathbb{E}_{x'} f(x') \|^2 \\
&= \mathrm{Var}(f(x)|y).
\end{aligned}
\tag{14}
$$

$\square$

## A.3 PROOF OF PROPOSITION 4.7

**Proposition A.6.** *Under Assumptions 4.5, & 4.6, by minimizing the InfoNCE loss, we can conclude that the conditional variance term vanishes, i.e.,*

$$
\mathrm{Var}(f(x) \mid y) = 0.
\tag{15}
$$

*Proof.* Consider any $\mathcal{T}$-connected sample $x_i, x_j$. Accoding to the definition of $\mathcal{T}$-connectivity, there exist $t_i, t_j \in \mathcal{T}$ such that $t_i(x) = t_j(x)$. When perfect alignment holds as in Assumption 4.6, we will have $f(x_i) = f(t_i(x_i))$ and $f(x_j) = f(t_j(x_j))$. Combining with $t_i(x_i) = t_j(x_j)$, we have $f(x_i) = f(x_j)$. That is, any $\mathcal{T}$-connected pair has the same representation. Then, in the augmentation subgraph $\mathcal{G}_k$ that is connected according to Assumption 4.5, there exists a path for any pair of samples $\hat{x}_i, \hat{x}_j \in \mathcal{G}_k$ where any two adjacent samples are $\mathcal{T}$-connected. As a result, $\hat{x}_i$ and $\hat{x}_j$ will also have the same representation by applying $\mathcal{T}$-connectivity recursively. At last, all samples in $\mathcal{G}_k$ will have the same representation and the intra-class variance vanishes. $\square$

## A.4 PROOF OF THEOREM 4.8

**Theorem A.7** (Guarantees for the optimal encoder). *If Assumption 4.1, 4.5 & 4.6 hold and $f$ is $L$-smooth, then, for the minimizer $f^\star = \arg \min \mathcal{L}_{\mathrm{NCE}}(f)$, its classification risk can be upper and lower bounded by its contrastive risk as*

$$
\mathcal{L}_{\mathrm{NCE}}(f^\star) - \mathcal{O}\left(M^{-1/2}\right) \leq \mathcal{L}_{\mathrm{CE}}^\mu(f^\star) + \log(M/K) \leq \mathcal{L}_{\mathrm{NCE}}(f^\star) + \mathcal{O}\left(M^{-1/2}\right).
\tag{16}
$$

*Proof.* A direct combination of Theorem 4.2 and 4.7 will give us the above two-sided bounds. $\square$

## B    Generalized Guarantees under Weak Alignment

In Section 4.2, we have shown that with perfect alignment (Assumption 4.6), the variance terms in the bounds of Theorem 4.2 can be minimized to zero, and consequently, the upper and lower bounds can be asymptotically closed. Nevertheless, in practice, due to the constraint of hypothesis class $\mathcal{F}$ and optimization algorithms, we typically cannot achieve the exact minimizer, *i.e.,* a perfect degree of alignment. This motivates us to consider a less restrictive setting, namely the $\varepsilon$-weak alignment assumption, where the alignment error could be as large as $\varepsilon$.

**Definition B.1** (Weak Alignment). *A mapping $f$ satisfies $\varepsilon$-weak alignment if $\forall \ x, x^+ \sim p(x, x^+), \|f(x) - f(x^+)\| \leq \varepsilon$.*

For any $\varepsilon$-weak alignment $f \in \mathcal{F}$, we have the following bounds on its downstream risk.

**Theorem B.2** (Guarantees under weak alignment). *If Assumption 4.1, 4.5 hold, then $\forall f \in \mathcal{F}$ satisfying $\varepsilon$-weak alignment, its classification risk can be upper and lower bounded by its contrastive risk as*

$$\mathcal{L}_{\mathrm{NCE}}(f) - D\varepsilon - \frac{1}{2}D^2\varepsilon^2 - \mathcal{O}\left(M^{-1/2}\right)$$
$$\leq \mathcal{L}_{\mathrm{CE}}^{\mu}(f) + \log(M/K) \leq \mathcal{L}_{\mathrm{NCE}}(f) + D\varepsilon + \mathcal{O}\left(M^{-1/2}\right), \tag{17}$$

*where $D$ denotes the maximal diameter of the intra-class augmentation graphs $\{\mathcal{G}_k, k = 1, \dots, K\}$ and $m$ denotes the output dimension of the encoder $f$.*

In this way, we extend the guarantees developed for optimal encoders (Theorem 4.8) to even non-minimizers $f \in \mathcal{F}$ as long as it could align the positive samples within error $\varepsilon$.

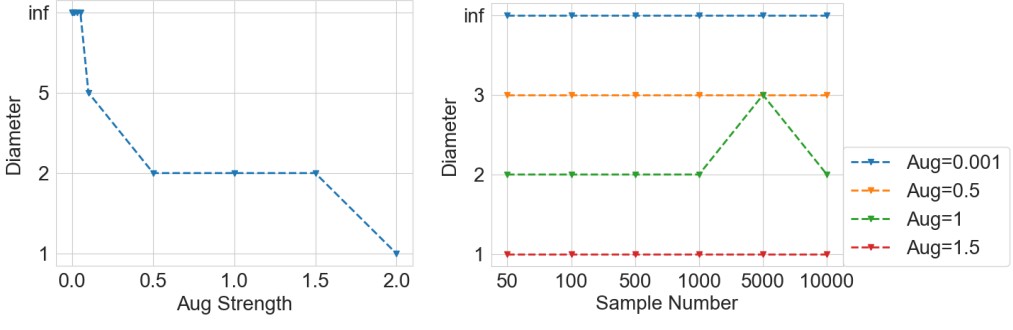

(a) Diameter $D$ *v.s.* augmentation strength $r$.     (b) Diameter $D$ *v.s.* the number of samples.

Figure 8: Evaluation of the maximal diameter $D$ as a function of different augmentation strength (a) and different number of samples (b) on the synthetic data in Section 5.1.

**Empirical Verification.** Besides the alignment error, we could notice this relaxation also introduces the dependence on an additional parameter $D$, the maximal diameter of the intra-class augmentation graphs. As shown in Figure 8, when the augmentation is very weak, the intra-class graph is not connected and the diameter is $\infty$. Then, by applying stronger augmentations, $D$ will become smaller and smaller, and finally converge to 1 (fully connected). Besides, increasing the number of samples, ranging from 50 to 10,000, does not have a large impact on $D$ in practice. Given these facts, we could reasonably assume that $D$ is bounded and has a relatively small value with properly chosen augmentations. As a result, with a bounded diameter $D$, a small alignment error $\varepsilon$ will guarantee a small generalization gap between the upstream and downstream tasks. This generalizes Theorem 4.8 by quantifying the generalization gap under weak alignment.

*Proof.* Consider any pair of samples $(x, x')$ from the same class $y$, and the positive sample of $x$ as $x^+$. As intra-class connectivity holds, $x$ and $x'$ are connected, and the maximal length of the path from $x$ to $x'$ is $D$. Therefore, under the $\varepsilon$-weak alignment that

$$\forall x, x^+ \sim p(x, x^+), \|f(x) - f(x^+)\| \leq \varepsilon, \tag{18}$$

we can bound the representation distance between $x$ and $x'$ by the triangular inequality

$$\|f(x) - f(x')\| \leq D \sup_{p(x, x^+ | y \sim p(x, x^+))} \|f(x) - f(x_+)\| \leq D\varepsilon. \tag{19}$$

With the inequality above, we can bound the variance terms in Theorem 4.2. In particular, the conditional variance can be bounded as

$$
\begin{aligned}
&\mathrm{Var}(f(x) \mid y)\\
=&\mathbb{E}_{p(y)}\mathbb{E}_{p(x|y)}\|f(x) - \mathbb{E}_{x'}f(x')\|^2\\
=&\mathbb{E}_{p(y)}\mathbb{E}_{p(x|y)}\|\mathbb{E}_{x'}f(x) - f(x')\|^2\\
\leq&\mathbb{E}_{p(y)}\mathbb{E}_{p(x|y)}\mathbb{E}_{p(x'|y)}\|f(x) - f(x')\|^2\\
\leq&\mathbb{E}_{p(y)}\max_{x,x'\sim p(x|y)}\|f(x) - f(x')\|^2\\
\overset{(1)}{\leq}&\mathbb{E}_{p(y)}D^2\varepsilon^2 = D^2\varepsilon^2
\end{aligned}
\tag{20}
$$

where (1) follows Eq. 19. At last, we can bound the variance items in Theorem 4.2 with Eq. 20, arrive at the desired bounds

$$
\begin{aligned}
&\mathcal{L}_{\mathrm{NCE}}(f) - D\varepsilon - \frac{1}{2}D^2\varepsilon^2 - \mathcal{O}\left(M^{-1/2}\right)\\
\leq&\mathcal{L}_{\mathrm{CE}}^{\mu}(f) + \log(M/K) \leq \mathcal{L}_{\mathrm{NCE}}(f) + D\varepsilon + \mathcal{O}\left(M^{-1/2}\right),
\end{aligned}
$$

which conclude our proof. $\qquad\square$

## C   ADDITIONAL EMPIRICAL EVIDENCE

### C.1   FURTHER EVALUATION OF ARC METRIC

In the main text, we study the effect of different strength of RandomResizedCrop on the downstream accuracy as our proposed metrics (ACR and ARC), which help verify our theory. Nevertheless, in practice, the augmentations adopted in contrastive learning is composed of a list of different kinds of augmentations. Therefore, in this part, we further study the effect of other types of data augmentations, and we show that our ARC metric is also effective for evaluating not only other kinds of data augmentations, but also their composed ones.

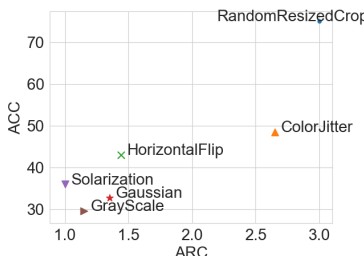

Figure 9: Downstream accuracy (ACC) *v.s.* Average Relative Confusion (ARC) for different types of augmentations in SimCLR on CIFAR-10.

**Comparing different kinds of augmentations.** We begin by comparing the four kinds of data augmentations adopted in SimCLR (Chen et al., 2020a): RandomResizedCrop, ColorJitter, Grayscale, etc. For a fair comparison, we apply each one *alone* for contrastive learning, and evaluate both the downstream accuracy and ARC. From Figure 11, we can conclude that among the six kinds of augmentations, RandomResizedCrop is the most important augmentation, and ColorJitter is the second. The rest of them are less powerful, as they cannot even learn useful features by themselves. We can also see that our ARC metric aligns well with the downstream accuracy for different kinds of augmentations.

**Comparing ColorJitter with different strength.** Based on the observation above, as we have discussed RandomResizedCrop in Section 5.2, we now choose ColorJitter, the second important augmentation, as another kind of augmentation for the study of different augmentation strength. Specifically, we study the four parameters of brightness, contrast, saturation, and hue, where a large value corresponds a large degree of augmentation. Note that we also adopt the default augmentations in SimCLR while only changing the parameters of ColorJitter (different to the setup in Figure 9). As shown in Figure 10, there is also a reverse-U curve like that in RandomResizedCrop, and the

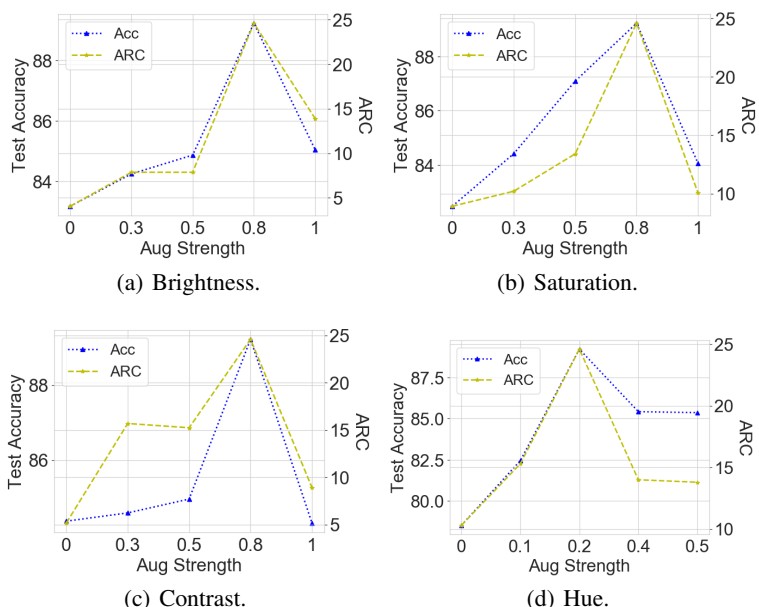

Figure 10: Average Relative Confusion (ARC) *v.s.* downstream accuracy with different augmentation strength on four different kinds of color jittering operations.

sweet spot is usually achieved with 0.8, which corresponds to the default of choice in SimCLR (which is selected with exhausted hyperparameter search). Meanwhile, our ARC metric still aligns well with the downstream accuracy for different strength of different kinds of color jittering, which demonstrates its wide applicability.

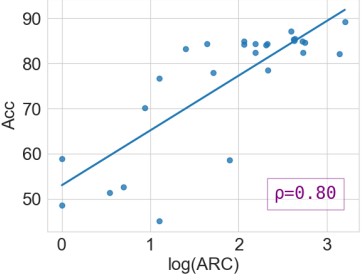

Figure 11: Downstream accuracy (ACC) *v.s.* the logarithm of Average Relative Confusion (ARC) on a composition of RandomResizedCrop and ColorJitter with different strength. Experiments are conducted on CIFAR-10 with SimCLR.

**Comparing composed augmentations.** In the above discussion, we focus on the effect of a single kind of augmentations. Here, we show that our ARC metric is still effective for evaluating the composition of different augmentations. Notably, it is hard to define a metric of augmentation strength in this case, as the effect of different augmentations could be nested. Nevertheless, we can still draw a "ACC - log(ARC)" plot to show the correlation between the downstream accuracy (ACC) and our ARC metric, where each point denotes a model trained with randomly selected parameters of RandomResizedCrop *and* ColorJitter. As shown in Figure 11, we can see there is indeed a strong correlation between the two metrics, with a Pearson correlation coefficient $\rho = 0.80$. Therefore, our metric can be used for selecting different kinds of augmentations as well as their compositions in an unsupervised fashion.

## C.2 VISUALIZATION OF AUGMENTATION GRAPH

For a more intuitive and practical understanding of our augmentation overlap theory developed in Section 4, we visualize of the augmentation graphs on both synthetic data (Section 5.1) and real-world data (Section 5.2).

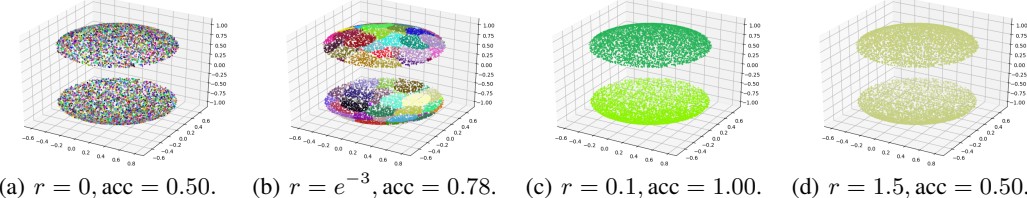

(a) $r = 0$, acc $= 0.50$.    (b) $r = e^{-3}$, acc $= 0.78$.    (c) $r = 0.1$, acc $= 1.00$.    (d) $r = 1.5$, acc $= 0.50$.

Figure 12: Visualization of the augmentation graph with different augmentation strength $r$ on the synthetic data described in Section 5.1. Each color denotes a connected component. The corresponding t-SNE visualization and test accuracy (of contrastive learning) can be found in Figure 4.

**Synthetic data.** Following the setting of experiments in Section 5.1, we construct the adjacent matrix of different samples, calculate its connected components, and visualize it in Figure 12 with different colors. It shows that when there is no augmentation, *i.e.,* $r = 0$, each sample is a connected component alone, and the number of connected components is the same as the number of samples $N$. As we increase the augmentation strength, samples will be connected together through the augmented views. In particular, when $r = 0.1$, the whole intra-class samples are connected while inter-class samples are separated, which exactly satisfy our assumptions on intra-class connectivity and label consistency, respectively. Therefore, this is the perfect overlap as desired, and indeed, as shown in Figure 5.1, contrastive learning on it obtains 100% test accuracy. When we keep increasing the augmentation strength to be as large as $1.5$, inter-class samples also become connected and inseparable, leading to a random guess in test accuracy (50%). This shows that the relationship between the augmentation graph and the downstream performance aligns well with our augmentation overlap theory.

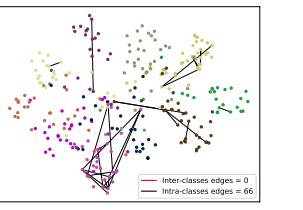 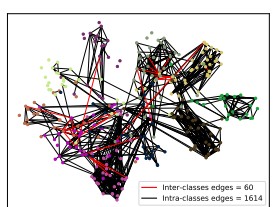 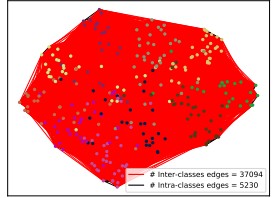

(a) Under-overlap augmentation graph (r=0.01, acc=0.25).    (b) Proper overlap augmentation graph (r=0.92, acc=0.75).    (c) Over-overlap augmentation graph (r=1.96, acc=0.29).

Figure 13: The augmentation graph of CIFAR-10 with different strength $r$ of RandomResizedCrop as in Section 5.2. We choose a random subset of test images, randomly augment each one for 20 times. Then, we calculate the sample distance in the representation space as in prior work like FID (Heusel et al., 2017), and draw edges for image pairs whose smallest view distance is below a small threshold. Afterwards, we visualize the samples with t-SNE and color intra-class edges in **black** and inter-class edges in **red** and report their frequencies.

**Real-world data.** For the ease of analysis, our augmentation overlap theory adopts a simplified scenario by assuming label consistency (Assumption 4.1) and intra-class connectivity (Assumption 4.5), and we have verified their feasibility on the synthetic data. In comparison, these assumptions cannot hold exactly on real-world data as the chosen augmentations could be sub-optimal. Nevertheless, as shown in the augmentation graphs of CIFAR-10 (Figure 13), our assumptions could still approximately hold: with a properly chosen augmentation strength, the inter-class connections will be much less frequent than intra-class connections: 96.4% edges are intra-class edges. Further considering the continuity property and extrapolation ability of deep neural networks, these approximate conditions could still achieve close performance to the optimal performance guaranteed under the

exact conditions. Besides, we also have similar conclusions for the under-overlap and over-overlap scenarios: 1) the lack of enough augmentations produces only a few edges in the augmentation graph, as a result, even though all edges are intra-class edges, the downstream performance is still poor (25% test accuracy); 2) too strong augmentations instead produce too many inter-class edges (87.6%), which laso leads to poor downstream accuracy (29%). This highlights that our assumptions on label consistency and intra-class connectivity are indeed effective guidelines for the designing of contrastive methods.

## D  THEORETICAL CHARACTERIZATION OF AUGMENTATION STRENGTH

Following the setting in Section 5.1, we can take the radius $r$ as a notation of augmentation strength, and analyze its effect on the connectivity of the corresponding augmentation graph.

**Theorem D.1.** *For $N$ random samples taken from a class, while gradually increasing the augmentation strength $r$, we have the following results.*

(a) **Under-overlap.** *When $0 \leq r \leq r_1 = \frac{[(d/2)!]^{\frac{1}{d}}}{\sqrt{\pi}} (\frac{1}{d})! (\frac{S}{N-1})^{\frac{1}{d}} [1 - \frac{1/d+1/d^2}{2(N-1)} + O(\frac{1}{(N-1)^2})]$, where $r_1$ is the minimal distance between $N$ samples, all samples (vertices) in the augmentation graph will be isolated. As a result, the learned features could be totally random as in Proposition 3.1. Instead, if $r \geq r_1$, there are at least two intra-class samples are $\mathcal{T}$-connected and enjoy the same representation.*

(b) **Perfect overlap.** *When $r \geq r_2 = \frac{[(d/2)!]^{\frac{1}{d}}}{\sqrt{\pi}} \frac{(N-2+1/d)!}{(N-2)!} (\frac{S}{N-1})^{\frac{1}{d}} [1 - \frac{1/d+1/d^2}{2(N-1)} + O(\frac{1}{(N-1)^2})]$, where $r_2$ is the maximal distance between $N$ samples, all samples in the augmentation graph will be $\mathcal{T}$-connected. As a result, the classwise connectivity in Assumption 4.5 will be guaranteed.*

(c) **Over-overlap.** *When $0 \leq r < r_3 = \frac{1}{2} \min_{i,j} \|c_i - c_j\| - 1$, where $r_3$ is the (asymptotic) minimal distance between samples from different classes, the label consistency is guaranteed. Otherwise, when the augmentation is too large, e.g., $r > r_3$, there will be inter-class augmentation overlap and Assumption 4.1 not longer holds.*

In the theorem above, we show that the proper augmentation strength is a function of the number of samples $N$ and the input dimensions $d$. In particular, for each $x$, as $N$ increases, there will be more natural examples and we only need a smaller $r$ to obtain an overlap sample. Instead, as $d$ increases, due to the curse of dimensionality, there will be less samples within the same distance, thus it requires a larger $r$.

Nevertheless, we actually only need the augmentation sub-graph $G_k$ to be connected, instead of being fully connected as in Theorem D.1 (b). While the connectivity is hard to analyze in the finite sample scenario ($N < \infty$), we have the following asymptotic property as $N \to \infty$.

**Theorem D.2.** *For $N$ uniformly distributed samples defined in $\mathbb{R}^d$ as above, we denote the minimal augmentation strength needed for connectivity as a function of $N$: $c_N = \inf\{r > 0 : G_k^{(r)} \text{ is connected}\}$, and the minimal augmentation strength needed for avoiding isolated points as a function of $N$: $d_N = \inf\{r > 0 : \text{every vertex at least has a neighbour}\}$. $V_u$ is the volume of unit hyperball. Then we have the following asymptotic result:*

$$\forall d \geq 2, \; \lim_{N \to \infty} \left( c_N^d \frac{N^2}{\log N} \right) = \lim_{N \to \infty} \left( d_N^d \frac{N^2}{\log N} \right) = 2 \frac{(1-1/d)S}{V_u}. \tag{21}$$

From the theorem we can see that $c_N^d$ decreases in the order of $\Theta\left( \sqrt[d]{\frac{\log N}{N^2}} \right)$ as $N \to \infty$. First, this result is aligned with the empirical finding that self-supervised learning can benefit more from large scale dataset (Chen et al., 2020b). Second, it also indicate a curse of dimensionality that the required augmentation strength is exponentially large.

### D.1  PROOF OF THEOREM D.1

*Proof.* From definition and notation in section 4. We can construct an augmentation Graph $\mathcal{G}(\mathcal{D}, \mathcal{T})$ given N random samples. We define $D_k$ as the distance from a random point to its k-th nearest

neighbour. Percus & Martin (1998) discuss $D_k$ in random grpah and give the estimation of that:

$$D_k \approx \frac{[(d/2)!]^{\frac{1}{d}}}{\sqrt{\pi}} \frac{(k-1+1/d)!}{(k-1)!} (\frac{S}{N-1})^{\frac{1}{d}} [1 - \frac{1/d+1/d^2}{2(N-1)} + O(\frac{1}{(N-1)^2})] \quad (22)$$

where d is the dimension of hypersphere and N is the number of random points. When $r < D_1$ there is no edge in the graph. So the class is separated. When $r > D_{N-1}$, any pair of vertexes have an edge between them,so the graph is full connected. $\square$

### D.2 PROOF OF THEOREM D.2

*Proof.* Denote

$$c_N = \inf\{r_i > 0 : G_N(V, E, r_i) \text{is connected}\}. \quad (23)$$

With Theorem 1.1 from Penrose (1999) and features are uniformly distributed in the surface of unit hypersphere, $V_u$ denotes to the volume of unit hypershpere

$$\lim_{N \to \infty} (c_N^d \frac{N^2}{\log N}) = 2\frac{(1-\frac{1}{d})S}{V_u}, d \geq 2 \quad (24)$$

$\exists N_0$ when $N > N_0$, and augmentation strength is larger than $(\frac{2(d-1)S \log N}{2N^2 V_u d})^{\frac{1}{d}} + \epsilon_1$, the graph is connected,i.e the class is overlapped.
Then we want specify the case the class will be depart begin with some concepts in graph theory. The largest nearest-neighbor link: For a give edge distance x and for each i= 1,...,n,let

$$\deg U_{N,j} = \sum_{1 \leq j \neq k \leq N} 1_{\{\|U_j - U_k\| \leq r_i\}} \quad (25)$$

to be the degree of the vertex $U_j$ in the random graph $G_N(V, E, r_i)$, and let

$$\delta_N(r_i) = \min\{\deg U_{m,1}(x), ..., \deg U_{N,N}(x)\} \quad (26)$$

be the minimum vertex degree.Define the largest nearest-neighbor link, the smallest edge distance for which each vertex has at least one neighbor

$$d_N = \inf\{r_i : \delta_N(r_i) \geq 1\} \quad (27)$$

With Theorem 1.2 from Penrose (1999),

$$\limsup_{N \to \infty} (d_N^d \frac{N^2}{\log N}) = 2\frac{(1-\frac{1}{d})S}{V_u}, d \geq 2 \quad (28)$$

$\exists N_0'$ when $N > N_0'$, and augmentation strength is less than $(\frac{2(d-1)S \log N}{2N^2 V_u d})^{\frac{1}{d}} - \epsilon_2$, there will be at least 1 isolated point which is not connected to any other point,i.e the class is departed.
Thus $\exists N_1 = \max(N_0, N_0')$,when $N > N_1$, if augmentation strength is larger than $(\frac{2(d-1)S \log N}{2N^2 V_u d})^{\frac{1}{d}} + \epsilon_1$, the graph is connected, if augmentation strength is less than $(\frac{2(d-1)S \log N}{2N^2 V_u d})^{\frac{1}{d}} - \epsilon_2$, there will be at least 1 isolated point. $\square$

## E ADDITIONAL EXPERIMENTAL DETAILS

### E.1 SIMULATION ON RANDOM AUGMENTATION GRAPH

Following our setting in Section 5.1, we consider a binary classification task with InfoNCE loss. We generate data from two uniform distribution on a unit ball $\mathbb{S}^2$ in the 3-dimensional space. One center is $(0, 0, 1)$ and another is $(0, 0, -1)$. The area of both parts are 1. We take 5000 samples as train set and 1000 samples as test set. For the encoder class $\mathcal{F}$, we use a single-hidden-layer neural network with softmax activation, and we use InfoNCE loss to optimize it.

### E.2 EXPERIMENTS ON REAL-WORLD DATASETS

To better understand and verify our theorem, we conduct experiments on real-world datasets, including CIFAR-10, CIFAR-100 and STL-10. We use SimCLR (Chen et al., 2020a) and BYOL Grill et al. (2020) as our training framework and use ResNet18 as our network. For CIFAR-10 and CIFAR-100, we adopt $C = 10$ augmentations for each image, and search neural neighbors in the entire augmented dataset. For STL-10, we adopt $C = 6$ due to its relatively large size.

