# OpenReview forum: "Chaos is a Ladder: A New Theoretical Understanding of Contrastive Learning via Augmentation Overlap"
_ICLR.cc/2022/Conference — ICLR 2022 Poster_

### Official Review · Reviewer_bTLa · 2021-10-26

**Correctness:** 4
**Technical Novelty And Significance:** 2
**Empirical Novelty And Significance:** 2
**Recommendation:** 6
**Confidence:** 4

**Main Review:**

Strengths:

The authors provided a new understanding of contrastive learning from the perspective of data augmentation for intra-class samples. Moreover, to evaluate the effect of data augmentation a quantitative analysis is provided along with a new metric.

############

Weaknesses:

. Theorem 4.2: For the downstream classification, the loss is upper and lower bounded in terms of the L_NCE loss. The authors provided comparison with Saunshi et al. (2019) from the technical perspective. Is there any intuitive explanation on how to evaluate the classification performance in terms of contrastive learning (loss)?

. Assumption 4.5 (intra-class connectivity): This assumption is strong. Without the label information, it seems impossible to derive such augmentation set. Please add discussion on the practicality of this assumption, and show an example on some datasets if possible.

. Proposition 4.7: Based on the proof provided in the appendix the conclusion not only relies on the existence of such augmentation set (Assumption 4.5), but also that such augmentation should be applied to intra-class samples, ie, t_i(x_i) = t_j(x_j). This kind of operation is impractical without the label information. Please add comment on that.

. In the experiments, RandomResizedCrop is used to illustrate the relationship between Aug Strength and ACC(ARC). The best performance for different datasets all achieves at Aug Strength = 0.92. Any comments on that? eg., in terms of data augmentation for intra-class samples at Aug Strength = 0.92?

. In practice, there are different kinds of data augmentation, eg, flipping, rotation, and scaling. The authors only showed results on RandomResizedCrop. Can you show results for other data augmentation types? Do you have similar conclusion as that for RandomResizedCrop?

. Different data augmentation types are often used together in practice (eg, randomly pick two augmentations from the augmentation set for the raw image). Then how to apply the proposed analysis in such practical case? In particular, how to measure the Aug Strength?

. The authors emphasized the importance of the data augmentation design for intra-class samples (ie, perfect overlapping). 1) The study on applying the analysis to existing contrastive learning algorithms is, however, preliminary (only with RandomResizedCrop). 2) Based on the proposed analysis how to find the sweet spot of data augmentation for contrastive learning is crucial, but this is not discussed.



**Summary Of The Paper:**

The authors provided a new understanding of contrastive learning from the perspective of data augmentation for intra-class samples. In particular, the authors proposed to understand the role of data augmentation as to create certain ``chaos'' between intra-class samples so to encourage the clustering of intra-class samples and also the learning of class-separated representations. Additionally, a new metric ARC is proposed to evaluate the downstream performance. The conclusion is validated via both synthetic and real-world datasets.

**Summary Of The Review:**

The idea of understanding contrastive learning from the perspective of data augmentation for intra-class samples is interesting. However, 1) some key assumption for the analysis is too strong; 2) the analysis on the existing contrastive learning algorithms is preliminary and needs more work; and 3) the authors emphasized the importance of finding the sweet spot of data augmentation (ie, perfect overlapping). But how to achieve that in practice is not discussed.

---

> ### Author Response · Authors · 2021-11-18
> **Response to Reviewer bTLa (3/3)**
>
> **Q4.** The best performance for different datasets all achieves at Aug Strength = 0.92. Any comments on that?
>
> **A4.** This is mainly a coincidence with a sparse choice of the augmentation strength. We have tried a few augmentation strengths near 0.92, and find the optimal value is 0.9 for CIFAR-10 and 1.22 for CIFAR-100. Thus, the optimal augmentation strength is indeed different among different datasets and contrastive methods, though they could be relatively close.
>
> ---
> **Q5.** The authors only showed results on RandomResizedCrop. Do you have similar conclusion for other data augmentation types as that for RandomResizedCrop?
>
> **A5.** Following your suggestions, we have added some new results in **Appendix B.1**: 1) in **Figure 10**, we show that our ARC metric still aligns well with the downstream accuracy for evaluating **different types of augmentations** adopted in SimCLR; 2) in **Figure 11**, we further show that our ARC metric is still effective when evaluating another type of augmentation, **ColorJitter with different augmentation strength**. We refer to the paper for more detailed discussions.
>
> ---
> **Q6.**  Different data augmentation types are often used together in practice (eg, randomly pick two augmentations from the augmentation set for the raw image). Then how to apply the proposed analysis in such practical case? In particular, how to measure the Aug Strength?
>
> **A6.** Following your suggestion, in our newly added **Appendix B.1**, we show that our proposed ARC metric is still effective for evaluating the composition of different augmentations. Notably, it is hard to define a metric of augmentation strength in this case, as the effect of different augmentations could be nested. Nevertheless, we can still draw a "ACC - log(ARC)" plot to show the correlation between the downstream accuracy (ACC) and the logarithm of our ARC metric. Specifically, each point in the plot denotes a model trained with randomly selected parameters of **RandomResizedCrop *and* ColorJitter**. As shown in **Figure 12**, we can see there is indeed a strong correlation between the two metrics, with a Pearson correlation coefficient $\rho=0.80$. Therefore, our metric can be used for selecting different kinds of augmentations as well as their compositions in an unsupervised fashion.
>
> ---
> **Q7.** How to find the sweet spot of data augmentation for contrastive learning is crucial, but this is not discussed.
>
> **A7.** Our proposed ARC metric is exactly designed for finding the sweet spot of data augmentations in practice. Its design is motivated by our theory for finding a proper degree of support overlapping (measured by the ACR score). As discussed in **Section 5.2**, we notice that the relative change of ACR scores indicates the sweet spot of augmentation strength, which motivates us to design the ARC metric as a surrogate metric of downstream performance. As shown in **Figure 7 & 8**, ARC indeed **aligns well with the sweet spot** of classification accuracy, and our newly added experiments in **Appendix B.1** (discussed in **A5, A6**) further demonstrated its effectiveness with **different kinds of augmentations as well as composed ones**. Through these thorough experiments, we have shown that ARC is indeed an effective metric for finding the sweet spot of data augmentation.
>
> ---
> Thanks for your careful reading and detailed review. Hope our explanations and extended empirical justifications could address your concern. Please let us know if you have additional questions.

---

> ### Author Response · Authors · 2021-11-18
> **Response to Reviewer bTLa (2/3)**
>
> (continuing A2)
>
> ### 2) Empirical evidence on synthetic and real-world data
>
> For ease of analysis, our overlapping theory adopts a simplified scenario by assuming label consistency and intra-class connectivity, and we have verified their feasibility on the synthetic data. In particular, in **Figure 13** in the newly added **Appendix B.2**, we show that there exists a perfect augmentation that will connect all intra-class samples while separating inter-class samples. In this way, we show that we can indeed find an optimal augmentation strength by **using only the domain knowledge of the data distribution** but **not the labels of every sample**.
>
> Instead, on real-world data, these assumptions cannot hold exactly as the chosen augmentations could be suboptimal. Even so, we can empirically show the augmentation graph is still very close to our assumptions, and we can achieve reasonably well performance with these approximate conditions. Specifically, we visualize the augmentation graphs of CIFAR-10 in the newly added **Figure 14**. We can see that our assumptions indeed hold approximately: with a properly chosen augmentation strength, the inter-class connections will be much less frequent than intra-class connections (**96.4\% edges are intra-class edges**), and the intra-class edges connect most intra-class samples together. Further, considering **the continuity property and extrapolation ability** of deep neural networks, these approximate conditions could still achieve close performance to the optimal performance guaranteed under the exact conditions.
>
> Besides, we also have similar conclusions for the **under-overlapping and over-overlapping** scenarios:
> 1) the lack of enough augmentations produces only a few edges in the augmentation graph, as a result, even though all edges are intra-class edges, the downstream performance is still poor (25\% test accuracy);
> 2) too strong augmentations instead produce too many inter-class edges (87.6\%), which leads to poor downstream accuracy (29\%).
>
> This highlights that our assumptions on label consistency and intra-class connectivity are indeed effective guidelines for the designing of contrastive methods.
>
> ### 3) A technical comparison with conditional independence [1,2]
>
> As we mentioned in Introduction (Page 1), the Conditional Independence (CI) assumption adopted [1,2] on the positive samples is so strong that it is almost equivalent to having the label information.
> >  ... if the conditional independence is satisfied, the unsupervised task will become as informative as the supervised task, making this discussion almost unnecessary.
>
> In fact, CI corresponds exactly to how labels are used in supervised contrastive learning [3] (drawing two independent samples from the same class as positive samples). In comparison, our theory stresses that **positive samples are not conditionally independent**, and only need them to be intra-class connected (Assumption 4.5).
>
> To see this difference, let us assume that CI indeed holds: because we could draw two conditionally independent positive samples, the intra-class augmentation graph will become a **complete** graph, which is unrealistic. While in our theory, we only assume the graph to be **connected** (Assumption 4.5), which is much weaker than completeness. Thus, our assumptions are indeed much weaker than CI, and thus much weaker than the supervised setting.
>
> Considering the three aspects above, we believe that we have shown our intra-class connectivity assumption is fundamentally different to knowing the labels of all samples in the supervised setting.
>
>
> [1] Saunshi et al., A theoretical analysis of contrastive unsupervised representation learning. ICML 2019.
>
> [2] Lee et al. Predicting what you already know helps: Provable self-supervised learning. arxiv 2020.
>
> [3] Khosla et al. Supervised contrastive learning. NeurIPS 2020.
>
>
> ---
> **Q3.** Proposition 4.7: The conclusion not only relies on the existence of such augmentation set (Assumption 4.5), but also that such augmentation should be applied to intra-class samples, i.e., $t_i(x_i) = t_j(x_j)$. This kind of operation is impractical without the label information.
>
> **A3.** First, we note that we **do not require additional assumptions** other than Assumption 4.5 & 4.6 for proving Proposition 4.7. We recall that the definition of $\mathcal{T}$-connectivity (Def 4.3) already states that two samples are $\mathcal{T}$-connected if  two nodes have overlapping augmentations, i.e., $t_i(x_i) = t_j(x_j)$. Therefore, as we assume intra-class connectivity (Assumption 4.5), we have already assumed the existence of such augmentations, and do not rely on additional assumptions. As for its practicality and relationship with the label information, please refer to the discussion above (**A2**).

---

> ### Author Response · Authors · 2021-11-18
> **Response to Reviewer bTLa (1/3)**
>
>
> We thank Reviewer bTLa for the detailed review and constructive comments, although there might be some misunderstandings of our theory. We will address your concerns in the following points.
>
> ---
> **Q1.** Theorem 4.2: Is there any intuitive explanation on how to evaluate the classification performance in terms of contrastive learning (loss)?
>
> **A1.** In Theorem 4.2, we showed the relationship between the contrastive loss (InfoNCE) and the classification loss (CE). An important conclusion that we made is that, although we could establish upper and lower bounds on the classification loss in terms of the contrastive loss, there are still some gaps between the two losses, measured by the variance terms in Eq. 3. Importantly, these gaps are unavoidable without further assumptions, as shown in the counterexample in Proposition 3.1.
>
> Therefore, in order to evaluate the downstream performance with our bounds, we need to calculate **contrastive loss + the variance terms**. In practice, the contrastive loss is simply the empirical InfoNCE loss (Eq. 1), and the variance terms could also be estimated with a finite number of samples. With these two estimates, we can calculate the upper and the lower bounds of the downstream performance following Theorem 4.2 (Eq. 3).
>
> ---
> **Q2.** Assumption 4.5 (intra-class connectivity): This assumption is strong. Without the label information, it seems impossible to derive such augmentation set. Please add discussion on the practicality of this assumption, and show an example on some datasets if possible.
>
> **A2.** We respectively disagree, and we must note that Assumption 4.5 is fundamentally ***not* equivalent to** knowing labels. We argue this point below from three different aspects.
>
> ### 1) Domain knowledge $\neq$ labels
>
> We highlight that our assumption on support overlapping describes the property between input images themselves. In fact, to achieve proper support overlapping (intra-class connectivity), we usually inject a certain degree of **domain knowledge** into the design of data augmentations [1]. For example, for an object-centric dataset like ImageNet, augmentations chosen by SimCLR are class-irrelevant operations (e.g., RandomResizedCrop, ColorJitter, HorizontalFlip, Grayscale, etc). In this work, our theory provides a rigorous understanding of **how these domain-specific augmentations work for contrastive learning**.
>
> In particular, our analysis is motivated by the observation in Figure 1b, that an aggressive cropping operation indeed could help **connect intra-class images even without knowing their specific labels**, e.g., cars with the same wheels and pens with the same nibs. Likewise, other SimCLR augmentations may also help connect similar but different raw samples through augmentations. As analyzed by our theory, the support overlapping created by augmentation is the key for contrastive learning to learn class-separated representations, without which contrastive learning is still useless as shown in Proposition 3.1. Therefore, our support overlapping theory only relies on the **domain knowledge** instead of **image labels**.
>
> [1] Lee et al. $i$-Mix: A Domain-Agnostic Strategy for Contrastive Representation Learning. 2021.
>
> (more on A2 below)

---

> ### Author Response · Authors · 2021-11-22
> **Need further clarification?**
>
> Thanks very much for your constructive and detailed comments. We have tried our best to address the concerns. Is there any unclear point that we should/could further clarify?

---

### Official Review · Reviewer_Pr6A · 2021-11-02

**Correctness:** 3
**Technical Novelty And Significance:** 3
**Empirical Novelty And Significance:** 3
**Recommendation:** 8
**Confidence:** 4

**Main Review:**

Strengths:
+ Theory considering both augmentation and alignment, without making too much assumptions.
+ Empirical verification on the niceness of a proper amount of augmentation.
+ The ACR and ARC metrics characterizes the interplay between augmentation and alignment, and are indicative of task performance.

Weaknesses:
+ The theoretical results are a bit weak. E.g., as pointed out in paper, Thm 4.8 only talks about the minimizer of the contrastive loss.

  Maybe this is unavoidable with the current set of augmentations. But can there be a version with the perfect alignment assumptions relaxed into approximate alignment? If so, it might be possible to talk about non-minimizers.

+ Proposition 3.1 is incorrect (but fixable I think).

  No finite samples can attain uniformity, because perfect alignment $\implies$ features are concentrated among finite number of vectors $\implies$ not a uniform distribution. The exact stated form is wrong, but I think some variants of it is true.

+ Figure 6.

  What is the experiment setting for this?

+ Sec. 5.1 "... And when $r$ is too large ($r=3$), ... "

  Is $r$ the geodesic distance on sphere or Euclidean distance in the ambient space? Either case, it is really large... (almost) containing the entire sphere! Is there not a milder augmentation that can also show the difference?


**Summary Of The Paper:**

The paper proposes a new theory for understanding contrastive representation learning. The novelty is the focus on the interplay between alignment and augmentation. Prior work has identified alignment as one of the factors of contrastive learning, but have not investigated how different types of augmentations may affect the learned embeddings. This work adds that missing piece. The results intuitively make sense, showing that proper amount of augmentation (that connects samples of the same class) has positive effect on downstream classification. Empirically, the authors verify that too weak or too strong augmentation harms performance. Based on observations, the authors define a metric on ratio of positive pairs among nearest (embedding) neighbors, and found the change of this metric throughout training positively correlate with performance.

**Summary Of The Review:**

The paper provides a theoretical analysis on the interplay between alignment and augmentations. Empirical experiments nicely complement the theory, and lead to interesting metrics that reveal the properties of this interplay. Overall the paper is also nicely written. While there is one slightly incorrect claim (which I think is fixable) and some places that would need clarification, I think the findings in this paper are valuable to the field. Thus, I recommend acceptance.

---

> ### Author Response · Authors · 2021-11-18
> **Response to Reviewer Pr6A**
>
> We thank Reviewer Pr6A for appreciating our work and highlighting its value to the field. Below, we address your concerns in the following points.
>
> ---
> **Q1.** Theorem 4.8 only talks about the minimizer of the contrastive loss. Can there be a version with the perfect alignment assumptions relaxed into approximate alignment? If so, it might be possible to talk about non-minimizers.
>
> **A1.** Following your advice, we provide a generalized version of our guarantees in **Appendix A** under weak alignment. In particular, we extend the guarantees developed for optimal encoders (Theorem 4.8) to even non-minimizers $f\in\mathcal{F}$ as long as it could align the positive samples within error $\varepsilon$. Please refer to the newly added **Appendix A** for more details and explanations.
>
> ---
> **Q2.** Proposition 3.1 is incorrect (but fixable I think).
> > No finite samples can attain uniformity, because perfect alignment $\Longrightarrow$ features are concentrated among finite number of vectors $\Longrightarrow$  not a uniform distribution. The exact stated form is wrong, but I think some variants of it is true.
>
> **A2.** Thanks for your careful reading, but we are afraid there is a subtle difference between our statement and your argument. Here, in Proposition 3.1, a direct implementation of the InfoNCE loss in Eq. 1 with $N$ samples $\{x_i\}_{i=1}^N$ would give the following empirical loss
>
> $$
> \mathcal{L}_{\rm NCE}(f)=\sum_i\left[f(x_i)^\top f(x^+_i)+\log\sum_j\exp(f(x_i)^\top f(x_j))\right]
> $$
>
> where we use the $N$ raw samples for the uniformity loss and draw $N$ additional positive samples $\{x_i^+\}_{i=1}^N$ for the alignment loss.
>
> Now let us take a look at our statement in Proposition 3.1:
> > consider the case when features $\{f(x_i)\}_{i=1}^N$ are randomly distributed in $\mathcal{S}^{m-1}$ with maximal uniformity while also satisfying $\forall x_i,x_i^+\sim p(x,x^+), f(x_i)=f(x^+_i)$
>
> Importantly, we only state that **the $N$ raw samples achieve maximal uniformity**, not including the additional positive samples $\{x_i^+\}_{i=1}^N$. If we do not include the positive samples in the uniformity loss (as we have done in the empirical loss above), the uniformity loss can indeed be maximized, and the minimum of InfoNCE can be obtained by our counterexample in Proposition 3.1. Instead, if the positive samples are also included, your argument is indeed right, and we cannot achieve perfect alignment and uniformity at the same time.
>
> In either case, our main message of Proposition 3.1 is clear: there will be poor solutions without support overlapping (this discussion is now included in the revised proof). The difference between your argument and ours only lies in the translation to an empirical loss. For ease of analysis, we tend to keep the current statement, as it is more concise. But we are open to further adjustments if you think it is still necessary.
>
> ---
> **Q3.** What is the experiment setting for Figure 6?
>
> **A3.** The setting is the same as Figures 7 & 8 that we mentioned on Page 9. We have now moved it ahead (before mentioning Figure 6) in the revision.
>
> ---
> **Q4.** Is $r$ the geodesic distance on sphere or Euclidean distance in the ambient space? Either case, $r=3$ is really large (Section 5.1). Is there not a milder augmentation that can also show the difference?
>
> **A4.** Here $r$ is the Euclidean distance, and we have replaced the $r=3$ case with $r=1.5$ instead (less than the diameter of the sphere) in **Figure 5** following your advice. The results and conclusions are still similar to the original one.
>
> ---
> Thanks for your insightful comments and hope our answers could address your concerns. Please let us know if you have additional questions.

---

> > ### Comment · Reviewer_Pr6A · 2021-11-22
> > **Reply to authors**
> >
> > Thanks for clarifying.
> >
> > I now think I understand what Proposition 3.1 is exactly referring to, but I think the current wording is confusing. In [1], "perfect uniformity" refers to the case where the feature distribution is exactly the uniform distribution on the hypersphere. However, in Proposition 3.1, it reads "Because we have perfect alignment and perfect uniformity" with finite samples, which makes perfect uniformity (as defined in [1]) impossible, whether positive pairs are included or not. Your reply above seems to interpret "perfect/maximal uniformity" as minimizer of the second term, but it is both inconsistent with prior work and not clear from the context. I think the wording should be best revised to not use "perfect uniformity" and to be clearer. For example, it could be worded like following:
> >
> > > ... are randomly distributed in $\mathcal{S}^{m−1}$ with maximal uniformity (i.e., minimizing 2nd term of Eqn. (2)), while also satisfying ....
> >  Because of these two properties, the InfoNCE loss achieves its minimum. However, ....
> >
> > [1] Wang and Isola., Understanding Contrastive Representation Learning through Alignment and Uniformity on the Hypersphere, 2020.

---

> > > ### Author Response · Authors · 2021-11-23
> > > **Thanks for your suggestions**
> > >
> > > We do agree that your wording is a more clear way to put Prop 3.1. We have revised the paper according to your suggestions.
> > >
> > > Thanks for appreciating our feedbacks and for your kind advice!

---

### Official Review · Reviewer_qd5j · 2021-11-03

**Correctness:** 4
**Technical Novelty And Significance:** 3
**Empirical Novelty And Significance:** 3
**Recommendation:** 8
**Confidence:** 3

**Main Review:**

In this exploration of data augmentation, much emphasis has been placed in the concept of "augmentation strength" - but what about the choices of augmentations themselves?  Can we perhaps use the ARC metric to evaluate, compare, and select data augmentation schemes themselves?  Separately, can the ARC metric be used to guide the selection of data augmentation parameters?  For example, for an arbitrary given augmentation scheme we can calculate the parameters that would maximize the ARC metric in an unsupervised way - then would applying those augmentations lead to comparable performance across different *choices* of augmentation strategies?  In other words, is the ARC metric a strong-enough metric that supercedes the selection of data augmentation strategies?  I would like to see more thought, analysis, and application regarding this new metric to fully convince me of its value and uses.

Additionally, to bridge the synthetic scenario and real data, I would like to see an augmentation graph of real augmented images, drawn with T-connections (where T can even be 1), and perhaps varied over different strength parameters.  I think there is definitely a gap between the authors' theoretical proposals/scenarios and that of actual natural data that can be closed with extra effort.  For example, the authors only mention one augmentation scheme to measure augmentation strength in real-world datasets, the RandomResizedCrop operator, and only evaluate it using their proposed metric.  Lastly, the reference section appears rather sparse, given the massive catalogue of work (including theoretical) surrounding contrastive learning.

Some typos:
- "alone cannot guarantee to learn class-discriminative..." should be "alone cannot guarantee the learning of class-discriminative..."
- "Comparing to Saunshi..., while ours only..." should be "Compared to Saunshi..., ours only" (page 6, Section 4.2).
- "and the surrogate could complete its mission..." should be " and the surrogate can complete its mission..." (page 7, Section 4.3)
- "different augmentation strength affects" should be "different augmentation strengths affect" (page 9, Section 6).
- "We take 500 sample as...For the encoder class , we..." should be "We take 500 samples as...For the encoder class, we" (page 16, Section D.1)

**Summary Of The Paper:**

The current leading theory of what contrastive losses are doing and why they work interprets contrastive learning as balancing alignment with uniformity, as proposed in [2].  This paper seeks to augment that understanding of contrastive learning using a new perspective, focusing on the role of data augmentation.  It is well-known that contrastive learning techniques are highly sensitive to the data augmentation schemes used, most notably discussed in [1].  In this work, the authors interpret augmentation as a way to connect different intra-class images together.  Then, the contrastive loss is seen as a way to gradually cluster intra-class samples together by aligning augmented views, producing representations that are class-separated even in feature space.

On top of introducing a new lens with which to understand contrastive learning, the authors also provide proofs on performance guarantees, as well as a new evaluation metric.  The metric is inspired by their augmentation-oriented understanding, and was also found to align well with downstream performance.

The authors provide a scenario where alignment and uniformity are satisfied, but fails to translate well to downstream classification accuracy.  This suggests to them that the instance discrimination task alone cannot guarantee the learning of class-discriminative features that would enable better downstream classification, and directs their attention to the other important component of contrastive-learning to help explain the story: augmentation.  They then build off the analytical work of [3] to prove guarantees for the downstream performance with a relaxed assumption.


[1] Chen et al., A Simple Framework for Contrastive Learning of Visual Representations, 2021.

[2] Wang and Isola., Understanding Contrastive Representation Learning through Alignment and Uniformity on the Hypersphere, 2020.

[3] Saunshi et al., A theoretical analysis of contrastive unsupervised representation learning, 2019.


**Summary Of The Review:**

The authors expand our understanding of contrastive learning on top of the existing alignment and uniformity perspective, by studying the role of data augmentation.  They provide theoretical guarantees on downstream performance, and propose an interesting new metric that can be evaluated using only the given unsupervised data.  Overall I think this is a strong submission, and would recommend an accept.

---

> ### Author Response · Authors · 2021-11-18
> **Response to Reviewer qd5j (2/2)**
>
> **Q5.**  The authors only mention one augmentation scheme to measure augmentation strength in real-world datasets, the RandomResizedCrop operator, and only evaluate it using their proposed metric.
>
> **A5.** As discussed above (**A1, A2, A3**), we have included more discussions using other kinds of augmentations in **Appendix B.2**, where our metric still aligns well with the practice across different kinds of augmentations.
>
> ---
> **Q6.** Lastly, the reference section appears rather sparse, given the massive catalogue of work (including theoretical) surrounding contrastive learning.
>
> **A6.** Following your suggestions, we have added more references in **Section 2 & 4.1**.
>
> ----
> Thanks for your encouraging comments, and hope we have addressed your concerns with the added experiments. Please let us know if you have additional questions.

---

> ### Author Response · Authors · 2021-11-18
> **Response to Reviewer qd5j (1/2)**
>
>
> We thank Reviewer qd5j for appreciating the novelty and solidness of our work. We have fixed the typos that you mentioned. Now we address your main concerns in the following points.
>
> ---
> **Q1.**  Can we perhaps use the ARC metric to evaluate, compare, and select data augmentation schemes themselves?
>
> **A1.** Following your suggestions, we add a new subsection **Appendix B.1** to evaluate different augmentation schemes. In particular, we compare **six** augmentation schemes adopted in SimCLR, e.g., RandomResizedCrop, ColorJitter, HorizontalFlip, Grayscale, Solarization, and Gaussian noise. For a fair comparison, we apply each of them alone for contrastive learning. From **Figure 10**, we can see that different augmentation schemes have very different effects on the downstream performance, where random cropping and color jittering are the two most important augmentations. In particular, we can notice that the ARC score aligns well with the downstream accuracy, so we can indeed use the ARC metric to evaluate, compare, and select data augmentation schemes themselves.
>
> ---
> **Q2.** Can the ARC metric be used to guide the selection of data augmentation parameters?
>
> **A2.** In Section 5.2, with **Figure 7 & 8**, we have shown that for the most important RandomResizedCrop augmentation, our ARC metric aligns well with the downstream accuracy across different augmentation parameters, particularly the sweet splot. This suggests that our ARC metric works well for selecting the data augmentation parameters.
>
> Besides, in **Appendix B.1**, we further study the second important augmentation scheme, **color jittering** in terms of the effect of its four parameters. Similar to that of RandomResizedCrop, as shown in **Figure 11**, our ARC also aligns well with the downstream accuracy across different parameters of color jittering, showing that ARC is good at selecting augmentation parameters as well.
>
> ---
> **Q3.** Is the ARC metric a strong-enough metric that supercedes the selection of data augmentation strategies?
>
> **A3.** With insights from the two experiments above, we think that our ARC could hopefully become a stronger indicator for the downstream performance without using labelled data. For a comprehensive study of their relationship, in **Appendix B.1**, we plot the "ACC - log(ARC)", where each point stands for the results with random parameters of **both RandomResizedCrop and ColorJitter**. We can see there is a strong correlation between the two metrics, suggesting our ARC is strong enough for selecting different kinds of augmentations as well as their compositions in an unsupervised fashion.
>
> ---
> **Q4.** I would like to see an augmentation graph of real augmented images that is varied over different strength parameters. I think there is definitely a gap between the authors' theoretical proposals/scenarios and that of actual natural data that can be closed with extra effort.
>
> **A4.**  We note that for ease of analysis, our overlapping theory considers a simplified scenario by assuming label consistency (Assumption 4.1) and intra-class connectivity (Assumption4.5). Following your suggestions, we provide more empirical evidence of our assumptions by visualizing the augmentation graph on both synthetic and real-world data in **Appendix B.2**.
>
> On the synthetic data, **Figure 13** shows that a clear improvement of connectivity as the augmentation strength grows, and the performance grows accordingly. In particular, when $r=0.1$, the whole intra-class samples are connected while inter-class samples are separated, which exactly satisfy our assumptions on intra-class connectivity and label consistency, respectively. After reaching this point, more aggressive augmentations will join samples from different classes together and lead to worse performance. This process aligns well with our analysis in **Section 5.1**.
>
> On real-world data, as you mentioned, these assumptions cannot hold exactly, as the chosen augmentations could be suboptimal. Nevertheless, we will see that these assumptions could still approximately hold. As shown in **Figure 14**, the default SimCLR augmentations produce much more intra-class edges than inter-class edges (96.4\% edges are intra-class edges), and the intra-class edges effectively connect most intra-class samples together, which is close to our assumptions. Further, considering the continuity property and extrapolation ability of deep neural networks, these approximate conditions could still achieve close performance to the optimal performance guaranteed under the exact conditions. We refer to more detailed discussions to **Appendix B.2**.

---

### Official Review · Reviewer_tWSB · 2021-11-04

**Correctness:** 3
**Technical Novelty And Significance:** 3
**Empirical Novelty And Significance:** 3
**Recommendation:** 6
**Confidence:** 4

**Main Review:**

**Strengths**:

- (S1) The problem being addressed is very relevant. Contrastive learning has enjoyed a lot of empirical success, and various works on theoretically understanding lack in one of many aspects when it comes to closeness to practice. This paper addresses issues with the theoretical assumptions and results in many prior work.

- (S2) Theorem 4.2 that upper bounds the downstream classification loss without conditional independence is new and interesting. The ACR metric that can select good augmentations using just unlabeled data is also an interesting finding.

- (S2) Various parts of the paper are accompanied with experiments (simulations and on standard datasets) to relate the theoretical analysis to practice

- (S3) Paper is clearly written and easy to follow

**Weaknesses**:

Here are many concerns about the theoretical assumptions and results that would help to have addressed by the authors.

- (W1) Assumption 4.6: One of the main concerns is the perfect alignment assumption, which *assumes* that the optimal solution $f^*$ of the NCE loss will satisfy $f^*(x) = f^*(x^+)$ for all positive samples $x$ and $x^+$. This seems like an unnatural assumption to make directly on the optimal solution, and is implicitly an assumption on the distribution of positive samples $p(x, x^+)$, since the optimal unit norm representations that minimizes that infoNCE loss depends strongly on this distribution. While some arguments for perfect alignment have been made in prior work [3], it is not clear whether that can be coherently imported here as an assumption. In fact, it is quite likely that the optimal infoNCE solution will not satisfy this assumption exactly for most joint distributions $p(x, x^+)$ (at the very least, a lot more justification is needed). This benign looking assumption undercuts the point that results here are shown under "less restrictive assumptions" compared to prior work, and it kind of trivializes the result in Theorem 4.8. **Note that the concern here is not just that the assumption is too strong or unrealistic (which is often unavoidable and acceptable), but that its not clear when the assumption can even be true and whether or not it is mathematically compatible with the rest of the setting.**.

- (W2) (Non-)vacuousness of bounds: I found Theorem 4.2 interesting since it can show a bound similar to the bound from [1] but without the conditional independence. One discussion I found missing is about how vacuous/non-vacuous the upper bound can be. Since the upper bound looks like $\mathcal{L}_{NCE}(f) - \log(M/K) + \sqrt{\text{Var}(f(x) | y)}$, it is not entirely clear whether this bound can ever be non-vacuous, i.e. are there cases where the sum of these terms can be very small. For e.g., in Theorem 4.8 where $\text{Var}(f(x) | y) = 0$, I can estimate a rough lower bound on this upper bound of $\log(M/K + M(1-1/K)/e^2) - \log(M/K) \approx \log(1 + (K-1)/e^2)$ which can be large for a large value of $K$. (here I used $\|f(x)\| = 1$). A discussion about the vacuousness (or not) of the bound can be critical in understanding whether the bound is indeed meaningful. A side note, given Theorem 4.2 and Proposition 4.7, Theorem 4.8 just seems like a corollary rather than a theorem.

- (W3) The result in [2] does not need conditional independence kind of assumption and in fact does analyze a more general case, albeit for a different spectral version of the contrastive loss. In particular, Assumption 4.1 from this paper will lead to $\alpha=0$ from that paper, and Assumption 4.5 from this paper will lead to reasonably high value for the Dirichlet conductance $\rho_{K}$ that shows up in their bound. Given that their results for spectral contrastive learning hold for the setting being considered in this paper, it is worth making a more detailed comparison to that paper.


**Other comments and questions**

- Section 5.1 seems to have some potentially interesting hypersphere example to demonstrate many of the points, but I thought it was not discussed enough in the main paper. It would help to give a short and clear summary of the results in Section B in the main paper.

- Some statements made in the paper deserve much more justification or could be toned down. E.g. "the class collision terms that are incompressible in Saunshi et al. (2019) now disappear in our bounds by adopting the InfoNCE loss, which also explains why InfoNCE performs better in practice": this does not really seem like an explanation for why InfoNCE performs better in practice, it is a weak justification at best. "increasing M indeed leads to a lower approximation error and helps close the gap" this is not clear since $\mathcal{L}_{NCE}(f)$ also depends on $M$.

- The setting for Proposition 3.1 is not described clearly, with regards to what kind of augmentation distributions (overlapping or not) does it hold for. I can only guess that it is for the case where they don't overlap for any pair of inputs, so it is not applicable when Assumption 4.5 is satisfied for example. Some clarification on this would be appreciated.

- Assumption 4.1 says that the conditional label distribution $p(y|x) = p(y|x^+)$ matches for positive samples $x$ and $x^+$. However this assumption is invoked in many places to say that inputs from different classes do not have overlapping support of augmentations and that the label is deterministic given $x$ or $x^+$, e.g. "Besides, because proper data augmentation will not cause inter-class support overlap (Assumption 4.1)" on page 6. Perhaps this assumption needs to be modified appropriately, or may be a separate assumption is needed about augmentation distributions not overlapping between inputs from different classes.

- The ACR metric makes sense and it is interesting that it helps in practice. but connection to theory is weaker than it is made out to be. After all the theory only talks about the overlap between augmentation distributions, but nothing about the nearest neighbors w.r.t. randomly initialized network features or the learned features.

- Will help to explain what $f_j(x)$ means in Theorem 4.2; seems like it means that $j^{th}$ coordinate of $f(x)$.

- Proposition 4.7 should only be true for $f^*$ and not all $f$.

- Is "chaos" used as a technical term? If so, any citation for its prior usage would be useful to include.

- Missing citations: [4,5] theoretically analyze contrastive learning for downstream task, [6] reports inverted-U shaped curves as in Figures 7 and 8 in this paper.

[1] Arora et al., A theoretical analysis of contrastive unsupervised representation learning. 2019.

[2] HaoChen et al., Provable guarantees for self-supervised deep learning with spectral contrastive loss. 2021.

[3] Wang et al., Understanding contrastive representation learning through alignment and uniformity on the hypersphere. 2020.

[4] Tosh et al., Contrastive estimation reveals topic posterior information to linear models. 2020.

[5] Tosh et al., Contrastive learning, multi-view redundancy, and linear models. 2020.

[6] Tian et al., What Makes for Good Views for Contrastive Learning? 2020.

**Summary Of The Paper:**

This paper aims to provide theoretical understanding for contrastive learning where "similar pairs" of points $x$ and $x^+$ are encouraged to have similar representations through an InfoNCE inspired objective function. Some prior works show the benefit of learned representations for linearly classifying downstream classes, by making conditional independence like assumption on the similar pairs or positive samples, i.e. $x$ and $x^+$ are (approximately) conditionally independent given downstream label $y$. This work argues that these assumptions are quite strong for contrastive learning with data augmentations, and aims to show guarantees under the following weaker and more realistic assumption: support of augmentation distribution of different inputs from the same class overlap to form a "connected graph" of inputs within a class, whereas support of augmentations of inputs from different classes do not overlap. Lower and upper bounds using this and some other assumptions, connecting the downstream performance of representation function to the contrastive loss. Some simulation experiments are presented to support some aspects of the theoretical analysis.

Using the insights from the analysis, the paper proposes an "Average Confusion Ratio (ACR)" metric that can be used to predict the ranking of downstream performances of different augmentations **using only unlabeled data**. Experimental evidence is provided on CIFAR and STL datasets to verify the efficacy of this metric for some practical augmentations.


While there are some interesting aspects in the paper (especially the ACR metric), the theoretical analysis seems to have raised many questions and concerns that I have summarized below (details in main review).

- **Soundness of assumptions**: Assumption 4.6, which is crucial, seems questionable and may not be coherent or appropriate to make in this setting. More on this in point (W2) of main review

- **Deeper dive into theoretical results**: There is a lack of discussion about the (non-)vacuousness of the bounds in the main results Theorem 4.2 and 4.8, that puts the interpretation and significance of the result in question. More on this and related issues in point (W2) of main review.

- **Comparison to prior work**: The work of HaoChen et al. in particular is not adequately compared to, especially since some of the points being addressed here are covered through a different kind of analysis in that paper. More on this in point (W3) of main review.

**Summary Of The Review:**

The paper aims to provide some theoretical analysis for contrastive representation learning under weaker assumptions than prior work (like conditional independence) and has some interesting empirical findings about how performance of augmentations can be ranked using a metric that depends just on unlabeled data. While the general idea is nice, there are issues with the theoretical setup (as described in the main review), raising questions about the meaningful-ness of the assumptions and results. Furthermore the comparison to prior very relevant work is also inadequate. This leads me to assign a score of reject for the current version.

---

> ### Author Response · Authors · 2021-11-18
> **Response to Reviewer tWSB (4/4)**
>
> **Q5.** The class collision terms do not really seem like an explanation for why InfoNCE performs better in practice, it is a weak justification at best.
>
> **A5.** We agree with you that it could only help justify rather than (fully) explain the practice, so we have modified the statement from "which explains" to "which helps understand" following your advice.
>
>
> ---
> **Q6.** The setting for Proposition 3.1 is not described clearly, with regard to what kind of augmentation distributions (overlapping or not) does it hold for.
>
> **A6.** We note that Proposition 3.1 is only meant to establish **the existence of a counterexample**, while its clear characterization is delayed till the introduction of the concept $\mathcal{T}$-connectivity (Section 4.3 & 5). But indeed, we need to specify the overlapping condition to give a rigorous counterexample. Thereby, we have revised the proof in **Appendix C.1** to be more clear on this point.
>
> ---
> **Q7.** Assumption 4.1 needs to be modified on the deterministicity  of labels.
>
> **A7.** Indeed, we have implicitly assumed the labels of samples to be deterministic, and we have added it to Assumption 4.1 in the revision following your suggestions.
>
> ---
> **Q8.** The connection between the ACR metric and the theory is weak.
>
> **A8.** In fact, our ACR metric is designed as a stochastic approximation to **measure the degree of the support overlapping**. We note that a direct evaluation of the degree of support overlapping is to calculate **the average degree of the nodes (samples)** in the augmentation graph. However, to establish the augmentation graph, we need to evaluate the $\mathcal{T}$-connectivity between any pair of samples $(x, x')$, where we need to enumerate all possible augmented views of $(x, x')$ to decide whether they share a common view. With a large augmentation space $\mathcal{T}$, there could be an exponentially large amount of augmented views, making this calculation computationally prohibitive.
>
> In view of this problem, in ACR, we resort to an efficient stochastic estimation by randomly sampling a fixed number ($C$) of views for each sample. In this case, exact examination ($0/1$) of common views is inappropriate as we omit many views, so we instead measure $x$'s degree of support overlapping by counting the number of views from other samples among $x$'s $k$-nearest neighbors (over all randomly selected augmented views). Intuitively, if $x$'s support overlaps more with that of other samples, there will be a larger probability where views of other samples appear in its $k$-nearest neighbors. So this estimation is consistent in expectation.
>
> The calculation above will lead to a ratio in $[0,1]$, which is exactly our proposed confusion rate of $x$: $\operatorname{CR}_k(x)$. ARC further calculates the dataset-level degree of support overlapping by an average of $\operatorname{CR}_k(x)$. Therefore, we believe that our ACR is a reasonable and efficient measure for support overlapping that is centric in our theory.
>
> ---
> **Q9.** Explain what $f_j(x)$ means in Theorem 4.2.
>
> **A9.** Indeed, it means the $j$-th coordinate of $f(x)$. We have added this explanation in the revised version.
>
> ---
> **Q10.** Proposition 4.7 should only be true for $f^\star$ and not all $f$.
>
> **A10.** Yes, it should hold only for the minimizer $f^\star$ as we mentioned by saying that "by minimizing the InfoNCE loss". We have also fixed the notations in the revision to avoid possible confusion.
>
> ---
> **Q11.** Is "chaos" used as a technical term? If so, any citation for its prior usage would be useful to include.
>
> **A11.** No, it is just a metaphor of our assumptions on $\mathcal{T}$-connectivity. Globally, our assumption appears as the overlapping of sample support illustrated in Fig. 4, while locally, it appears as the confusion between different augmentations illustrated in Fig. 1b. The latter observation is the starting point of our analysis, and our theory hinges on the insight that such chaos (confusion) created by data augmentations is critical for contrastive learning to generalize. Therefore, we use "chaos is a ladder" as a metaphor for our theory.
>
> ---
> **Q11.** Missing citations.
>
> **A11.** Following your suggestions, we have added them in **Section 2**.
>
> ---
> Thanks again for your detailed comments, which are very helpful, and hope our response could address your concerns. Please let us know if you have additional questions.

---

> ### Author Response · Authors · 2021-11-18
> **Response to Reviewer tWSB (3/4)**
>
> (continuing A3)
>
> Nevertheless, we note that as we take different technical approaches (our from bridging the InfoNCE loss and the CE loss, while theirs from a matrix decomposition approach), **the two results do not imply each other**. First, ours provides both upper and lower bounds, while they only provide upper bounds. Second, as we highlight above, in either case, there are additional conditions that [1] requires achieving the same result while ours do not. In particular, there are several cases that their method fails to analyze while ours could:
> 1) when our intra-class connectivity holds and $\lfloor k/2 \rfloor =r$, we have $\rho_{\lfloor k/2\rfloor}=\rho_{r}=0$, which leads to a $\frac{0}{0}$ term in the upper bound of [1] that they fail to characterize. Instead, our bounds do not have a dependence on $k$, and we can still obtain a sharp bound in this case;
> 2) when $\alpha=0$, their bound seems to suggest that we can immediately obtain a sharp bound, but this is not true, as we show in the counterexample in Proposition 3.1: when there is no support overlapping, contrastive learning could still learn nothing at all, even if samples are class-separable. In fact, in [1], this extreme case corresponds to $\rho_{i}=0, \forall i \leq N$, where $N$ is the number of samples. As a result, their bound again becomes $\frac{0}{0}$, and they again fail to characterize this case. Instead, we show that this case is fundamentally different to the case above, and the two enjoy very different bounds: no augmentation could yield large variance terms while with intra-class connectivity we can minimize them.
>
> This discussion shows that our analysis characterizes the difference between the two important extreme cases, while [1] cannot tell them apart.
>
>
> Besides the technical differences discussed above, we also highlight the following methodological differences from theirs:
> - **Different objectives.** [1] develop a new spectral loss and establish guarantees only for their proposed objective, while ours establishes guarantees for the widely adopted InfoNCE loss and CE loss, making it more applicable to explain a range of representative CL methods, e.g., SimCLR, MoCo and many of their variants.
> - **Different perspectives.** With their new objective, [1] casts contrastive learning as a matrix decomposition problem and study the effect of the spectral properties of the augmentation graph on the decomposition error. Instead, ours starts with the typical understanding of contrastive learning, i.e., alignment + uniformity, and we further renew this understanding by highlighting the role of augmentation, which could provide more insights into the common understanding of contrastive learning.
> - **Different implications in practice.** Although inspiring, the matrix decomposition perspective is quite different from the common designing principles of "contrastive learning": aligning positive samples and separating negative samples. This gap could make their theory provide fewer insights on the practical designing of CL methods, e.g., the choice of positive and negative samples.  In comparison, we accompany each of our assumptions with empirical evidence in contrastive learning and verify our theory on both synthetic and real-world datasets. Moreover, our theory also motivates an unsupervised metric, ARC, for evaluating contrastive learning. This shows that our theory could provide concrete insights into the practice of contrastive learning.
>
> Following your suggestion, we have added a brief summary of this discussion in **Section 2**.
>
>
> [1] HaoChen et al., Provable guarantees for self-supervised deep learning with spectral contrastive loss. 2021.
>
> ---
> **Q4.** Section 5.1 seems to have some potentially interesting hypersphere example ... It would help to give a short and clear summary of the results in Section B in the main paper.
>
> **A4.** Indeed, we also like this part, and we have added a summary of the main conclusion in Appendix to **Section 5.1** following your suggestion. Specifically, the conclusion is two folds: the minimal $r$ for the graph to be connected will decrease as the number of samples $N$ increases, so a large dataset can bring better connectivity; meanwhile, $r$ will increase when the input dimension $d$ increase, and we need more samples with large-size inputs.

---

> ### Author Response · Authors · 2021-11-18
> **Response to Reviewer tWSB (2/4)**
>
> (continuing A1)
>
> **(Our extension)** Nevertheless, as you point out, perfect alignment is indeed too ideal and hard to achieve in practice, making the discussions a bit rough. In view of this, in the revision, we have added a new section, **Appendix A**, where we provide a generalized version of our guarantees under weak alignment and its empirical evidence. In particular, we establish our theory with a $\varepsilon$-**weak alignment assumption**, where the alignment loss could be minimized up to an error $\varepsilon$. In **Theorem A.2** (quoted below), we show that we could also obtain similar bounds up to an error term (related to $\varepsilon$) for weak alignment solutions.
>
>
> > **Definition A.1 (Weak alignment).** A mapping $f$ satisfies $\varepsilon$-weak alignment if $\forall\ x,x^+\sim p(x,x^+), \Vert f(x)-f(x^+)\Vert\leq\varepsilon$.
> >
> > **Theorem A.2 (Guarantees under weak alignment).** If Assumption 4.1, 4.2 hold, then $\forall f\in\mathcal{F}$ satisfying $\varepsilon$-weak alignment, its classification risk can be upper and lower bounded by its contrastive risk as
> >$$
> {L}_{{NCE}}(f)-\left(1+\frac{\sqrt{m}}{2}\right) \sqrt{D} \varepsilon-\mathcal{O}(M^{-1/2}) {\leq}
> $$
>
> >$$
> \mathcal{L}_{\mathrm{CE}}^{\mu}(f)+\log (M / K) \leq
> $$
>
> >$$
>  \mathcal{L}_{\mathrm{NCE}}(f)+\sqrt{D} \varepsilon+\mathcal{O}\left(M^{-1 / 2}\right)
> $$
> > (separated due to markdown rendering issue), where $D$ denotes the maximal diameter of the intra-class augmentation graphs $\{\mathcal{G}_k,k=1,\dots,K\}$ and $m$ denotes the output dimension of the encoder $f$.
>
> More details, proofs, and empirical evidence could be found in **Appendix A**.
>
>
> Reference:
>
> [1] Wang et al., Understanding contrastive representation learning through alignment and uniformity on the hypersphere. ICML 2020.
>
> [2] Saunshi et al., A theoretical analysis of contrastive unsupervised representation learning. ICML 2019.
>
> [3] Lee et al., Predicting what you already know helps: Provable self-supervised learning. arxiv 2020.
>
> [4] Nozawa and Sato, Understanding  negative  samples  in  instance  discriminative  self-supervised representation learning. NeurIPS 2021.
>
> [5] Ash et al., Investigating the role of negatives in contrastive representation learning. arxiv 2021.
>
> [6] Merad, About contrastive unsupervised epresentation learning for classification and its convergence. arxiv 2021.
>
> [7] Anonymous, Sharp Learning Bounds for Contrastive Unsupervised Representation Learning. [https://openreview.net/forum?id=tDirSp3pczB](https://openreview.net/forum?id=tDirSp3pczB)
>
> ---
> **Q2 (W2).**  Non-vacuousness of bounds. In Theorem 4.2,  the bounds scale with $\log(M/K)$, which can be large for a large value of $K$.
>
> **A2.** We are afraid that you have missed the explanation of this problem in Theorem 4.2 that appears in footnote 1 on Page 5, where we stated that $\log(M/K)$ is a negligible constant as it could be absorbed in the loss functions by replacing ``sum`` with ``mean`` in InfoNCE and CE.
>
> Actually, it is very necessary to do this correction as the scale of InfoNCE loss grows unboundedly with larger $M$ (Eq. 1) by itself, making it hard to match it to the CE loss (there must be an $M$ term when relating the two). Instead, if we both take the mean instead of sum (either over $K$ or $M$) in both objectives, **the two will be boiled down to the same scale**, and will be not be affected by the number of summarization items ($K$ or $M$).
>
> This $\log(M/K)$ term exactly **corresponds to this correction step, so it should *not* be considered as a part of the actual bounds**. Ignoring this term, in Theorem 4.8 and A.2, the bounds will become very close (and thus non-vacuous) when $M$ is large and the alignment error $\varepsilon$ is small.
>
> ---
> **Q3 (W3).** A more detailed comparison with Haochen et al. [1]
>
> **A3.** Thanks for pointing it out as [1] is indeed closely related to us. Haochen et al. also establish guarantees for contrastive learning by studying the augmentation graph, and obtain similar conclusions to us. For example, denote $k$ as the output dimension, $r$ as the number of ground-truth classes, $\rho$ as the Dirichlet conductance in [1], and $\alpha$ the prediction error, we can observe that
> 1) our label consistency (Assumption 4.1) implies $\alpha=0$ in [1], which eliminates the first term **(if $\rho_{\lfloor k/2\rfloor}$ is not zero)** in the upper bound of Theorem 3.7 in [1];
> 2) when our intra-class connectivity (Assumption 4.5) holds, in Theorem 3.7 in [1], their $\rho_{\lfloor k/2\rfloor}>0$ holds **when $k>\lceil 2r \rceil$**, which also implies a non-vacuous bound in Theorem 3.7.
>
> (more on A3 below)

---

> > ### Public Comment · ~Kento_Nozawa2 · 2021-11-18
> > **let me clarify our assumption to improve this work**
> >
> > Dear authors,
> >
> > Thank you for kindly replying to my comment.
> >
> > I'm afraid that our paper does not assume
> >
> > > conditional independence of the two positive samples _i.e._, $p(x, x^+ \mid y) = p(x \mid y)p(x
> > ^+ \mid y)$
> >
> > as in Sec. 3.2 in my understanding because we created a positive sample $x^+$ by applying data augmentation to $x$.
> >
> > Note that our bound can be vacuous when the number of classes and the number of negative samples are small as empirically shown in F.4's result paragraph due to relaxing the assumption by using the data augmentation.

---

> > > ### Author Response · Authors · 2021-11-19
> > > **Response to Kento Nozawa**
> > >
> > > Thanks for pointing it out. Indeed, your work does not assume conditional independence as in [1,2], and thus has a similar variance-related gap as in ours. We apologize for our oversight.
> > >
> > > The difference between our bound (Theorem 4.2) and your bound (Theorem 8 [3]) mainly lies in the treatment of the incompressible **class collision** term in [1]: our bounds improve the bounds of [1] by eliminating the class collision; while in your bound, the class collision error still exists. In fact, even if we also substitute a $\log(M)$ from your bound as done in our analysis (explained in **A2**), the class collision error still remains incompressible in the order of $O(1)$; while in ours it is $\mathcal{O}(M^{-1/2})$, which becomes $0$ as the number of negative samples $M\to\infty$, making our bounds even sharper.
> > >
> > > We have revised the discussion in Section 4.1 following your suggestions.
> > >
> > >
> > > Reference:
> > >
> > > [1] Saunshi et al., A theoretical analysis of contrastive unsupervised representation learning. ICML 2019.
> > >
> > > [2] Lee et al. Predicting what you already know helps: Provable self-supervised learning. arxiv 2020.
> > >
> > > [3] Nozawa and Sato. Understanding negative samples in instance discriminative self-supervised representation learning. NeurIPS 2021.

---

> ### Author Response · Authors · 2021-11-18
> **Response to Reviewer tWSB (1/4)**
>
>
> We thank Reviewer tWSB for careful reading and detailed comments of our work and appreciate its novelty, though there might be some misunderstandings of our results.
>
> We will first summarize and address the three main weakness points you mentioned and then respond to the other minor comments.
>
> ---
>
> **Q1 (W1)**. As for Assumption 4.6, the perfect alignment assumption, has the following weakness,
>
> 1) it is unnatural as the optimal InfoNCE solution will not satisfy this assumption exactly, and maybe it is not mathematically compatible with the rest of the setting;
> 2) it trivializes the result in Theorem 4.8 and undercuts the point that results here are shown under "less restrictive assumptions" compared to prior work.
>
> **A1.** **(Problem 1)** Our Assumption 4.6 is directly borrowed from Wang et al. [1] that assumes **perfect alignment & uniformity** and serves as a starting example of our work as in Section 3.2. In Proposition 3.1, we show that **even** these two ideal and simplified conditions cannot guarantee a good downstream performance, and our work is a **nontrivial** extension of [1] by noting the importance of the effect of augmentation strength. Therefore, the main focus and insight of this work lie in the intra-class connectivity theory, while as for the alignment mechanism, we directly import Wang et al's ideal setting **for a clear comparison**: with our assumption on $\mathcal{T}$-connectivity, perfect alignment can now lead to guaranteed downstream performance.
>
> As for **its effect and compatibility with other results of this work**, we introduce this assumption at the last of our theory exactly because it is quite irrelevant to our discussion before. In particular, the general generalization bounds in Theorem 4.2, one of our key results, **do not depend** on this assumption at all. Besides, the key assumption, intra-class connectivity (Assumption 4.5), is made in the input space and will not be affected by Assumption 4.6. Together with the label consistency assumption (Assumption 4.1), there certainly exists an ideal encoder (given enough model capacity) that maps all intra-class samples to the same representation as this encoder could minimize the alignment loss. So this assumption, although ideal, is indeed compatible with our theory.
>
> **(Problem 2)** Besides, **does adding this Assumption break our statement that our theory is less restrictive than prior work [2,3]? Actually NO.** In Theorem 4.2, we give general bounds on the downstream performance without the Conditional Independence (CI) assumption [1,2] and Assumption 4.6. Going through the proof of Theorem 4.2, we can find that if we also assume CI like [1,2], we can easily eliminate the variance term $\operatorname{Var}(f(x)|y)$, that is, **directly obtaining the upper bound in Theorem 4.8** (without mentioning Assumption 4.6 at all). In fact, this variance term is **merely a result of the absence of CI,** and our overlapping theory focuses on eliminating it **without CI**. Therefore, the assumption in prior work [2,3] is indeed an (extreme) special case of our analysis. Besides, our bounds also improve over [1] in other aspects, such as a sharper bound with more negative samples. We note that several prior works have been devoted to resolving this issue **under CI** [4,5,6] including a concurrent submission [7], while ours is the first to show this problem can be resolved in general cases **even without CI**. We have elaborated more on this point in the revision (**Section 4.1**).
>
> (more on A1 below)

---

> ### Author Response · Authors · 2021-11-22
> **Need further clarification?**
>
> Thanks very much for your constructive and detailed comments. We have tried our best to address the concerns. Is there any unclear point that we should/could further clarify?

---

> > ### Comment · Reviewer_tWSB · 2021-11-26
> > **Response**
> >
> > I thank the authors for their responses to all reviews and the revisions made. The new results and experiments certainly make the paper stronger, especially the result in Appendix A. I have raised my score to 6 based on these. I would still like to point out a few things that I felt were unaddressed in the revision and could be important.
> >
> > **About Assumption 4.6**
> >
> > - As far as I understand, Wang et al. does not assume alignment and uniformity as stated in the response, but proves it under some simplifying assumptions. The response also states that *"intra-class connectivity (Assumption 4.5), is made in the input space and will not be affected by Assumption 4.6"* which is technically incorrect. Note that the optimal $f^{*}$ is eventually a function of the input and augmentation distributions and thus the augmentation graph. So these are not assumptions that can be made independently of each other, and they could in fact even be conflicting. For instance when the graph for a particular class is a union of many dense subgraphs that are connected to each other by just an edge, it is plausible that the optimal solution might not want to make the representations for all edges (especially the ones that connect different subgraphs) "align" with each other, in the interest of being more "spread out". While it is fine to make assumptions to prove results, care needs to be taken to ensure (or at least discuss) whether or not the assumptions are compatible with each other.
> >
> > - I will clarify what I mean by Assumption 4.6 trivializes Theorem 4.8. If we already willing to make a strong assumption like 4.6, then combining it with Assumption 4.5 will trivially give Proposition 4.7 (through transitivity). This would imply that all inputs from the same class will be assigned the same representation, and if these are different for different classes and dimensionality is at least #classes, then the downstream linear classification error can be "trivially" shown to be almost 0. In other words, one wouldn't even need to go through any additional bound involving the contrastive loss as in Theorem 4.8.
> >
> > Overall I feel like the technical results presented in Section 4.2 are still a bit shaky. The new result in Appendix A that uses weak-alignment can at least alleviate some of these issues (modulo understanding how small $\epsilon$ can be expected to be) and perhaps the authors should consider presenting that in the main paper instead.
> >
> >
> > **About vacuousness of bound**
> >
> > I believe the authors misunderstood my concern. It was not about the $\log(M/K)$ term, which I understand the purpose of. But it was about the final upper bound (say the one is Equation 5) and whether or not it can be vacuous. To elaborate, the upper bound is basically $L_{CE}(f) \le L_{NCE}(f) - \log(M/K)$ (ignoroing the $M^{-1/2}$ term). Note that $f$ is assumed to have unit norm and this is also used in the proof. Furthermore in the setting of Theorem 4.8, $f$ is the same for all inputs from the same class. So using the (in)equalities $f(x)^{\top} f(x') = 1$ when $x$ and $x'$ are from the same class, and that $f(x)^{\top} f(x') \ge -1$ when they are from different classes and plugging that into Equation 1, the RHS from above can be shown to be lower bounded by the quantity I mentioned in my original review ($\approx \log(1 + (K-1)/e^2)$). This lower bound is somewhat instance independent and could be a bit worrying that the bound is always vacuous. Though I realized that it might be possible to let the representation norm to be some constant $B > 1$ (and suffering $B$ in front of the variance term) and have a non-vacuous bound. A discussion about this could be useful nonetheless.
> >
> >
> >
> > **Connection of ARC to theory**
> >
> > The authors argue that ARC is a "stochastic approximation" to degree of support overlap by saying that "if $x$'s support overlaps more with that of other samples, there will be a larger probability where views of other samples appear in its $k$-nearest neighbors". While this statement seems intuitively true, the converse might not be true, which is what "connection of ARC to theory" is roughly implying. Just because there is a nearest neighbor from another input does not mean that the supports necessarily overlap. So either a more rigorous justification for the connection to theory should be presented, or the connection should be described as weak. Regardless of this, I think that the results with ARC are interesting by themselves.
> >
> >
> > I strongly encourage the authors to consider the above technical points and fix parts of the paper accordingly, if needed.

---

> > > ### Author Response · Authors · 2021-11-27
> > > **Further Response to Reviewer tWSB (2/2)**
> > >
> > > **Q2.** About vacuousness of bound.
> > >
> > > **A2.** We are afraid that we might not fully understand your argument and your derived bound here. As far as we could understand, to maximize the upper bound, you want to bound the alignment of positive samples by $-1$  in the nominator, and bound the alignment of negative samples by $1$ in the denominator, in the InfoNCE loss (Eq. 1). Our derivation yields the following bound,
> > > $$
> > > \mathcal{L}^\mu_{\mathrm{CE}}(f)\leq \mathcal{L}^\mu_{\mathrm{NCE}}(f)-\log(M/K)+\mathcal{O}(M^{-1/2})\leq -\log\frac{1/e}{M\cdot e}-\log(M/K)+\mathcal{O}(M^{-1/2})=2+\log K+\mathcal{O}(M^{-1/2}).
> > > $$
> > > With a large $K$, indeed this bound will be large. But **this is merely a result of a large value of CE loss itself**. Recall that CE loss is
> > > $$
> > > \mathcal{L}^\mu_{\mathrm{CE}}(f)=E_{p(x, y)}\left[-f(x)^{\top} \mu_{y}+ \log\sum_{i=1}^{K} \exp \left(f(x)^{\top} \mu_{i}\right)\right]\leq 2+\log K,
> > > $$
> > > that will also grow in the rate of $\log K$ because it has $K$ terms in the $\operatorname{logsumexp}$ term. Therefore, as mentioned in our previous response (**A2**), we suggest that both $\log M$ and $\log K$ terms should not be considered in the actual bounds, and they should be substituted away when comparing the InfoNCE loss and the CE loss.
> > >
> > > ---
> > > **Q3.** Connection of ARC to theory.
> > > > While this statement seems intuitively true, the converse might not be true, which is what "connection of ARC to theory" is roughly implying.
> > >
> > > Indeed, for now, the connection between ARC and theory is intuitively justified but not rigorously proved. Nevertheless, we are also surprised to see how it works effectively in practice. Thanks for appreciating it, and we will continue to work on improving it in the future.
> > >
> > > ---
> > >
> > > Thanks again for your detailed comments and encouraging score. Hope our explanations could ease your concerns. Please let us know if there is more to clarify.

---

> > > ### Author Response · Authors · 2021-11-27
> > > **Further Response to Reviewer tWSB (1/2)**
> > >
> > >
> > > We thank Reviewer tWSB for appreciating our response. We will further elaborate our explanations to address your concerns point by point.
> > >
> > > ---
> > > **Q1.** About Assumption 4.6.
> > >
> > > **Point a**
> > >
> > > > As far as I understand, Wang et al. does not assume alignment and uniformity as stated in the response, but proves it under some simplifying assumptions.
> > >
> > > Reviewing Theorem 1 in Wang et al., we can see that they indeed assume perfect alignment (point 1) and perfect uniformity (point 2) to conclude that the two InfoNCE terms are minimized, respectively. As far as we could see, they do not prove the opposite direction that the minimizer of the InfoNCE loss could guarantee that the two properties hold.
> > >
> > > **Point b**
> > > > [the statement] "intra-class connectivity (Assumption 4.5), is made in the input space and will not be affected by Assumption 4.6" is technically incorrect.
> > >
> > > In this statement, we only want to claim that in Assumption 4.5, **different intra-class samples** have connected support in the input space. Instead, Assumption 4.6 describes whether **the positive views of the same example** can be aligned **in the representation space**. Therefore, either Assumption 4.6 holds or not, it will not affect Assumption 4.5.
> > >
> > > **Point c**
> > > > So these [Assumptions 4.5 & 4.6] are not assumptions that can be made independently of each other, and they could in fact even be conflicting.
> > >
> > > Thanks for your explanations, and we now get your point. Indeed, different augmentation graphs have a different effect on the InfoNCE minimizer, i.e., 4.5 will have some effects on 4.6. Nevertheless, we still want to note that although there are some dependencies (so are the alignment and uniformity assumptions in Wang et al.), the two assumptions have very different focuses, and does not imply each other.
> > > Assumption 4.5 focuses on the connectivity between **different intra-class samples**, while Assumption 4.6 focuses on the alignment of **the positive views of the same example**. In particular, we note that by perfect alignment, we (as well as Wang et al.) assume that the alignment loss can be minimized **no matter what the augmentation graph is**. If all nodes in the graph are isolated, the alignment can still work, but we might get meaningless representations, as in Proposition 3.1. Instead, if intra-class samples are connected, i.e., combined with Assumption 4.5, and the alignment loss works (by connecting positive views of the same example), we can provably get class-clustered representations (which is exactly why we need Proposition 4.7).
> > >
> > > **Point d**
> > > > For instance, ... it is plausible that the optimal solution might not want to ... "align" with each other, in the interest of being more "spread out".
> > >
> > > Indeed, this is an interesting example and makes sense. Nevertheless, we are afraid that the "spread-out" property could not be reflected even in the CE loss itself, as the minimizer of the CE loss will also align all intra-class samples together. As our main focus in this paper is to establish bounds between the InfoNCE and CE losses, we have focused more on this alignment/clustering property. As for the feature diversity property mentioned in your example, our discussions on the effect of graph diameter in Appendix A could have some correspondences to it. Specifically, the diameter of the graph in your example will generally be larger than its densely connected version. In our analysis, it will lead to a large bound on the intra-class feature variance, i.e., better feature diversity. More involved analysis is worth exploring in the future.
> > >
> > > **Point e**
> > > > ... Assumption 4.6 trivializes Theorem 4.8 ... The new result in Appendix A that uses weak-alignment can at least alleviate some of these issues.
> > >
> > > Thanks for your explanations and suggestions. Indeed, Proposition 4.7 is a straightforward corollary when combining 4.5 and 4.6. The message here is to note that good representations can only be achieved through combining the two assumptions, and either of them alone is not enough (discussed above). That being said, we also agree that Appendix A offers a more general and realistic discussion, and we will consider adding it to the main paper in the revision.

---

### Public Comment · ~Kento_Nozawa2 · 2021-11-09
**Related work**

Dear authors,

Let me point out a missing related paper. Our paper [1] also conducts a theoretical analysis on self-supervised representation learning, where positive pairs are created by using data augmentation to understand the number of negative samples, not data augmentation that was not our target. So the main target in this paper is different from ours, but I believe that our work might be related to this paper because both papers analyse InfoNCE based representation learning.

Best regards,

----

[1] Kento Nozawa & Issei Sato. Understanding Negative Samples in Instance Discriminative Self-supervised Representation Learning. In _NeurIPS_, 2021. arXiv:2102.06866v3 [cs.LG].

---

> ### Author Response · Authors · 2021-11-18
> **Response to Kento Nozawa**
>
> Dear Kento,
>
> Thanks for bringing up your interesting work and we have added it to our discussion.
>
> Best,
>
> Authors

---

### Author Response · Authors · 2021-11-18
**A Summary of Paper Updates**


We thank all reviewers for the constructive suggestions, which helps make this work more complete. Following their suggestions, we have fixed the typos and made the following major updates to the paper, including **4+ pages** in Appendix A & B to provide more justifications of our theory:
- **Section 2**: add more references and discussions.
- **Section 4.1**: 1) explicate the determinstic property of labels in Assumption 4.1; 2)  explain $f_j(x)$ in Theorem 4.2; 3) add references and discussions of other extensions of Saunshi et al. (2019).
- **Section 5.1**: add a summary of the theoretical results in Appendix C.
- **Section 5.2**: move the experimental setup forward.
- **(New!) Appendix A**: a generalized version of our guarantees under weak alignment and its empirical evidence.
- **(New!) Appendix B.1**: evaluating our ARC metric under other kinds of data augmentations as well as composed ones.
- **(New!) Appendix B.2**: visualization of our augmentation graph on both synthetic dataset and real-world dataset (CIFAR-10).
- **Appendix C**: add more details to the proof of Proposition 3.1.

Please refer to the paper for further details.

---

### Decision · Program_Chairs · 2022-01-20

**Decision:**

Accept (Poster)

**Comment:**

The paper under review provides a theoretical analysis for contrastive representation learning. The paper proposes a guarantee on the performance (specifically upper and lower bounds) without resorting to previously used conditional independence assumptions. Throughout, the theoretical results and assumptions are supported by experiments.

After a lively discussion, and after changes made to the paper in the revision stage, all four reviewers recommend this paper for acceptance.
- Reviewer tWSB appreciates that the paper makes weaker assumptions than prior work (i.e., not assuming conditional independence), but raises a number of serious concerns on the theoretical results: The review questions whether assumption 4.6 used in the theory can be true, and whether the bound is vacuousness. The authors argue that this assumption was used in prior work, point out that only some of their results rely on this assumption, and that the assumption is compatible with the theory. The response of the authors partly resolved the reviewers concern and the reviewer raised their score.
- Reviewer bTLa finds the idea of understanding contrastive learning for intra-class samples interesting, but finds some key assumptions too strong, a critique similar to that raised by reviewer tWSB. The authors responded and the reviewer increased their score, and mentioned that most concerns were addressed. The response partially resolved the reviewers concern, and the reviewer now also recommends acceptance.

I recommend to accept the paper. Understanding contrastive learning better is an important problem, and based on my own reading, I agree with the reviewers that the paper contributes to the understanding of contrastive learning. Two reviewers had concerns about unrealistic assumptions, but those have been largely resolved in the discussion.